# Robust proteome profiling of cysteine-reactive fragments using label-free chemoproteomics

George S. Biggs [1,2,3,8], Emma E. Cawood [1,4,8], Aini Vuorinen [1,2,3],
William J. McCarthy [2], Harry Wilders [1,5], Ioannis G. Riziotis[1,6],
Antonie J. van der Zouwen[2], Jonathan Pettinger [1], Luke Nightingale[6],
Peiling Chen[7], Andrew J. Powell [1], David House [1], Simon J. Boulton [4],
J. Mark Skehel [3], Katrin Rittinger [2] ✉ & Jacob T. Bush [1] ✉

Identifying pharmacological probes for human proteins represents a key opportunity to accelerate the discovery of new therapeutics. High-content screening approaches to expand the ligandable proteome offer the potential to expedite the discovery of novel chemical probes to study protein function. Screening libraries of reactive fragments by chemoproteomics offers a compelling approach to ligand discovery, however, optimising sample throughput, proteomic depth, and data reproducibility remains a key challenge. We report a versatile, label-free quantification proteomics platform for competitive profiling of cysteine-reactive fragments against the native proteome. This high-throughput platform combines SP4 plate-based sample preparation with rapid chromatographic gradients. Data-independent acquisition performed on a Bruker timsTOF Pro 2 consistently identified ~23,000 cysteine sites per run, with a total of ~32,000 cysteine sites profiled in HEK293T and Jurkat lysate. Crucially, this depth in cysteinome coverage is met with high data completeness, enabling robust identification of liganded proteins. In this study, 80 reactive fragments were screened in two cell lines identifying >400 ligand-protein interactions. Hits were validated through concentration-response experiments and the platform was utilised for hit expansion and live cell experiments. This label-free platform represents a significant step forward in high-throughput proteomics to evaluate ligandability of cysteines across the human proteome.

Small molecule probes offer powerful tools for the study of biological systems and can serve as starting points for the development of therapeutics[1]. The vast majority of human proteins lack such chemical tools, which hinders our ability to explore the function of these proteins in the context of health and disease[2,3]. The development of high-quality ligands across the human proteome is now recognised as a key objective to enable the functional studies of biological systems and to accelerate identification of therapeutic opportunities[4–6].

[1]Crick-GSK Biomedical LinkLabs, GSK, Gunnels Wood Road, Stevenage, Hertfordshire, UK. [2]Molecular Structure of Cell Signalling Laboratory, The Francis Crick Institute, London, UK. [3]Proteomics Science Technology Platform, The Francis Crick Institute, London, UK. [4]DSB Repair Metabolism Laboratory, The Francis Crick Institute, London, UK. [5]University of Strathclyde, Pure and Applied Chemistry, Glasgow, UK. [6]Software Engineering and AI, The Francis Crick Institute, London, UK. [7]GSK Chemical Biology, GSK, Collegeville, PA, USA. [8]These authors contributed equally: George S. Biggs, Emma E. Cawood. ✉e-mail: Katrin.Rittinger@crick.ac.uk; jacob.x.bush@gsk.com

Traditional methods for ligand and tool discovery, such as high-throughput screening of small molecules with individual proteins, are resource intensive, prohibiting their utility in expanding the liganded proteome. Furthermore, the study of purified and often truncated forms of proteins is a poor reflection of the interactions made by the full-length proteins in their native cellular environment. Novel methods to enhance the throughput, scope, and accessibility of ligand discovery, particularly in the context of the native proteome, will be essential for realising the ambition of the research community to discover a chemical probe for every expressed protein[2,3,7].

Mass spectrometry (MS)-based chemoproteomics methods are emerging as powerful approaches to expand the ligandable proteome, enabling the identification of small molecule-protein interactions in a cellular context. Small molecule ligands that act via an irreversible covalent mechanism are particularly suited to these studies, as the interactions are retained through sample preparation and mass spectrometry analysis, allowing for the robust detection of binding events from complex mixtures[8–10]. The covalent bond can also provide an increase in potency due to prolonged target engagement, leading to the discovery of probe molecules from relatively low molecular weight chemotypes. The benefits of covalent fragment-based ligand discovery can then be exploited by using modestly sized libraries ($\sim10^2$–$10^3$) of low molecular weight fragments (<300 Da), while still efficiently covering chemical space[11–13].

Chemoproteomics screening of covalent small molecules was pioneered using activity-based probes (ABPs) for the development of inhibitors against specific enzyme families, e.g., serine hydrolases and de-ubiquitinases[14–17]. Screening compounds in competition with ABPs enables excellent sensitivity and coverage within the targeted protein family, but does not inform on the proteome-wide activity of these small molecules. Recent studies have developed chemoproteomics methods with expanded proteome coverage by using family-agnostic probes for nucleophilic residues, e.g., cysteine and lysine[18–21]. In particular, hyperreactive iodoacetamide probes that enable enrichment and quantification of cysteine-containing peptides have been employed for competitive cysteinome profiling of electrophilic fragments by chemoproteomics[22,23].

Key challenges still exist in the development of chemoproteomics platforms for competitive profiling of large compound libraries. Sample preparation and analysis must have sufficient throughput to profile entire libraries of covalent compounds, and the analytical technique must deliver sufficient sensitivity to detect a significant portion of the expressed proteome. Additionally, excellent reproducibility and data completeness is required to enable robust hit calling and the generation of full matrix datasets, which are suited to the implementation of machine learning models to drive iterative library design. To date, methods have employed long MS-proteomics acquisition times ($\sim$3-h chromatographic gradients) and data-dependent acquisition (DDA) mass spectrometry. Improvements in throughput have been achieved by multiplexing using tandem mass tag (TMT) labelling, however these batch-based DDA analyses can give low data completeness and poor reproducibility when combining datasets, which, in addition to the costs associated with TMT reagents, limits their accessibility for large library profiling[24–26].

Here, we present a high-throughput label-free quantification chemoproteomics (HT-LFQ) platform for profiling covalent libraries against the cysteinome. The method offers sensitivity and cysteinome coverage that compares favourably with reported methods to date, and offers improved reproducibility and data completeness. Sample preparation is performed using a 96-well plate-based workflow, consisting of a low-cost protein clean-up method with no requirement for isotopic labelling. Sample analysis employs label-free quantification (LFQ) and data independent acquisition (DIA), which yields excellent sensitivity and reproducibility of peptide detection. Importantly, the fragmentation and analysis of all precursors in DIA affords improved

data completeness between experiments, which contrasts with more variable peptide identification in DDA[27–29]. Using our HT-LFQ chemoproteomics platform, we screen a library of 80 chloroacetamide fragments against the cysteinome in two cell lines (HEK293T and Jurkat), and identify ligands for over 400 cysteine sites, including a number of protein families of high interest to drug discovery. The high-throughput nature and reproducibility of our platform allows ready access to further hit characterisation, including concentration-response studies and interrogation of structure-activity relationships. Collectively, we demonstrate that our HT-LFQ platform represents a powerful methodology to enable efficient discovery of chemical tools for proteins from biological samples.

## Results

### A high-throughput, label-free chemoproteomics workflow

To develop a high-throughput label-free quantification DIA chemoproteomics platform capable of screening compound libraries against the cysteinome, we employed a competitive profiling strategy using a previously reported hyperreactive iodoacetamide desthiobiotin (IA-DTB) probe[18]. Cell lysates were treated with cysteine-reactive fragments, followed by treatment with IA-DTB to enrich cysteine-containing peptides. MS-based quantification of enriched peptides and comparison of these peptide intensities between fragment-treated and DMSO control samples enabled calculation of the fragment engagement of each cysteine, reported as a competition ratio (CR).

We developed a sample preparation protocol to ensure reproducible recovery of cysteine-containing peptides from cell lysates in 96-well plate-based format, without the need for isotopic labelling (Fig. 1a). Following treatment of lysates with IA-DTB, a plate-based sample clean-up was performed, employing solvent precipitation on glass beads followed by an on-bead tryptic digestion. This precipitation approach allows for consistent protein recovery alongside efficient removal of excess small molecules and detergents[30]. Finally, digested and desthiobiotin-modified peptides were captured on high-capacity neutravidin resin and recovered using mildly acidic aqueous/organic mixtures. In this workflow, samples remain in 96-well plate format from compound treatment through to mass spectrometer injection, and a single experimentalist can readily prepare 384 samples (4 plates) in 2–3 days. The plate-based and label-free nature of this sample preparation method provides the throughput and reproducibility required for efficient screening of large libraries of covalent fragments.

Liquid chromatography-mass spectrometry (LC-MS) data was acquired using an Evosep One coupled to a Bruker timsTOF Pro 2. On this mass spectrometer, cysteinome coverage is maximised by separating peptides based on four properties (ion mobility, retention time, mass-to-charge ratio and intensity) and employing DIA with parallel accumulation-serial fragmentation (PASEF) for improved precursor identification and quantification (Fig. 1b)[31]. Initially, a hybrid (DDA/DIA) spectral library of IA-DTB-modified precursors was built utilising off-line peptide fractionation and longer chromatographic gradients. Subsequently, treated samples were analysed using short (21-min/60 samples per day) chromatographic gradients with DIA methods, and raw data were searched against the hybrid reference library. Data analysis was performed by comparison of the peptide intensities in treated and control samples, along with calculation of key statistics and data filtering (as described in the experimental methods) to identify the most robust and selective interactions.

### HT-LFQ chemoproteomics yields deep, reproducible peptide detection

To probe the reproducibility and coverage of our platform, we prepared 16 control (DMSO) samples from HEK293T and Jurkat lysates (32 samples in total). From these samples, we identified ~35,000 cysteine-containing peptides using 21-min chromatographic gradients (Fig. 1c and Supplementary Fig. 1a). On average, ~23,000 cysteine-

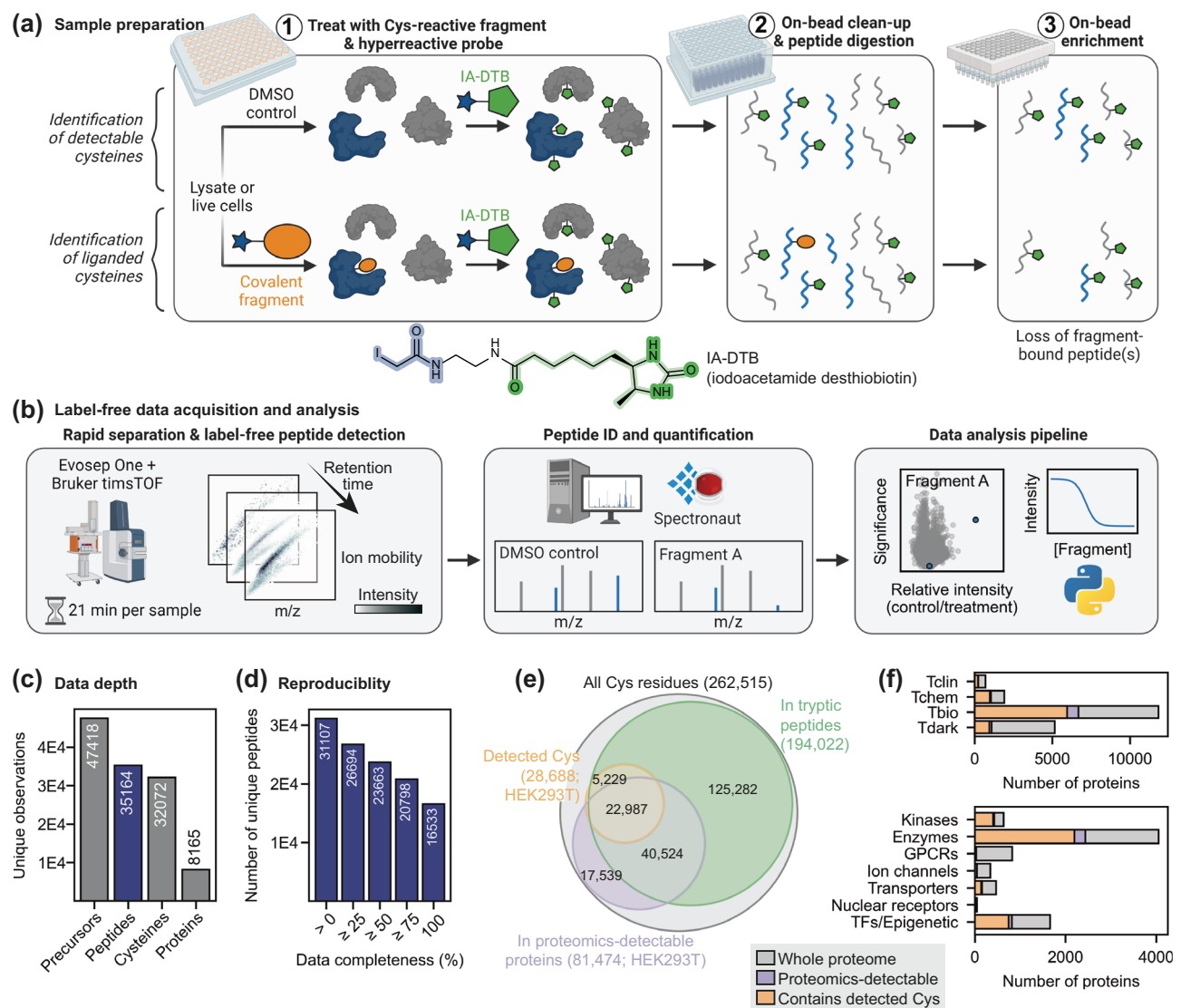

**Fig. 1 | Workflow and cysteinome coverage from our HT-LFQ chemoproteomic method. a** Schematic of our label-free sample preparation method that allows detection of cysteine-containing peptides from lysates or live cells using a hyper-reactive IA-DTB probe. **b** Data acquisition was performed using an Evosep One and Bruker timsTOF Pro 2, followed by identification and quantification of peptides using Spectronaut. Peptides that were bound by a covalent fragment are expected to show a reduced intensity in compound-treated samples, relative to control samples. **c** Our method allows detection of high numbers of peptides in HEK293T and Jurkat lysates, providing the opportunity to detect liganding events at over 30,000 cysteine residues from over 8000 proteins (*n* = 16 DMSO control samples from each lysate). **d** We see high data completeness of peptide detection, with two-thirds of peptides detected in ≥ 75% of samples, allowing more confident detection of binding events. Data shown here is from HEK293T lysate; see Supplementary

Fig. 1c for Jurkat data. **e** Our detection of cysteine residues from an individual cell lysate (HEK293T; orange) represents approximately ~40% coverage of residues that can be considered to be feasibly detectable, based on their location relative to tryptic cleavages sites (green) and the general detectability of proteins by global proteomics methods (purple), as well as the presence of disulfide bonds and post-translational modifications (see Supplementary Fig. 3). Tryptic peptides were classified as being detectable if they were 7–40 residues (not considering missed cleavages). **f** Distribution of proteins across protein families (top) and target development levels (bottom), as defined by the 'Illuminating the Druggable Genome' programme[3,7,35]. The colour scheme in this figure follows that used in (**e**). GPCRs: G protein-coupled receptors; TFs: transcription factors. Parts of both (**a**) and (**b**) were created in BioRender. Cawood, E. (2025) https://BioRender.com/ m32r739. Source data are provided as a Source Data file.

containing peptides were detected in each replicate, which matches the detection depth previously achieved with IA-DTB probes using 3-h gradients and isotopic labelling strategies (Supplementary Fig. 1b)[18,32]. Within each cell line, we observed high data completeness, with a median overlap of 82% between the peptides detected in any two replicates, and with two-thirds of all peptides being detected in over 75% of samples (Fig. 1d and Supplementary Fig. 1c, d). This high level of data completeness is a key advantage of label-free/DIA over TMT/DDA proteomics, and is essential for competitive profiling where inconsistent peptide detection can hinder comparison between control and treatment conditions[29]. Furthermore, excellent reproducibility of the peptide intensities was observed, with a median Pearson correlation

between replicates of 0.96 (Supplementary Fig. 1e), and a median coefficient of variation of 24.8% (Supplementary Fig. 1f, g). Comparison of the data acquired using HEK293T and Jurkat lysates revealed that the majority of detected peptides were observed in both cell lines, however inclusion of both cell lines allowed us to increase cysteine coverage by 10–15%, highlighting the potential to expand cysteinome coverage using cells of different biological origin (Supplementary Fig. 2).

## Features of the detected cysteines and proteins

The ~32,000 cysteine sites detected by our HT-LFQ platform come from over 8000 proteins, representing ~40% of the proteins in the

human proteome and 12% of all cysteine residues in the proteome. Various features of these detected cysteines and proteins were evaluated to assess their biological relevance and to rationalise factors that might affect detectability.

We rationalised that protein abundance and solvent accessibility may impact cysteine coverage. Protein abundance was approximated by the detection of proteins in global proteomics analysis of HEK293T cells, which revealed 80% of IA-DTB detected peptides arise from these abundant proteins, and cover ~30% of all cysteines in these proteins. Other features that are likely to lead to non-detection of cysteines include peptide physicochemical properties, involvement in disulfide bonds, and post-translational modifications (Fig. 1e and Supplementary Fig. 3)[33]. We additionally observed ~5000 cysteine sites from proteins we do not detect via global proteomics, likely due to the reduced complexity of chemoproteomics samples following the enrichment step.

The solvent accessibility of the cysteine residues detected in our platform was evaluated using previously reported prediction-aware part-sphere exposure (pPSE) values as a measure of solvent exposure[34]. We observed that the distribution of pPSE values for cysteines detected by our platform matches the distribution of the whole cysteinome (Supplementary Fig. 4), confirming that the probe is not biased towards the modification of highly exposed cysteine residues, consistent with previous analyses of IA-DTB coverage[32]. This is an important observation, as many functionally important cysteine residues lie within pockets or regions that are not fully solvent accessible. Engagement of more buried cysteine residues is expected due to protein dynamics, and, in some instances, modification of a single cysteine residue may lead to partial protein unfolding that increases the accessibility of additional residues.

To evaluate our coverage of the proteome with respect to protein function and prior knowledge of tractability, we referenced protein annotations from the 'Illuminating the Druggable Genome' initiative, which classifies proteins into one of four target development levels: 'Tclin', proteins that are already the targets of approved drugs; 'Tchem', proteins that have known small molecule ligands; 'Tbio', proteins that lack chemical tools but have well-studied biology; and 'Tdark', proteins for which very little information is known[3,7,35]. Using this protein classification, we detect ~7000 proteins in the Tbio and Tdark categories, highlighting the opportunity to identify probes for previously unliganded proteins (Fig. 1f). Furthermore, we see good coverage of proteins from families of strong interest to pharmaceutical drug development, such as kinases, as well as proteins from underrepresented families, such as transcription factors and epigenetic proteins[36]. Similarly, the families detected reveal good representation from nuclear proteins and enzymes. The observed underrepresentation of membrane-bound proteins (e.g., ion channels and GPCRs) is expected given their low solubility under the cell lysis conditions employed. Taken together, this analysis confirms that our HT-LFQ chemoproteomics platform offers the opportunity for largely unbiased profiling of a significant proportion of the proteome, including many proteins that currently lack chemical tools or small molecule drugs.

## Reactive fragment screening by HT-LFQ chemoproteomics

To test the applicability of the HT-LFQ platform for library screening, 80 chloroacetamide-functionalised fragments were screened against both the HEK293T and Jurkat proteomes. The library was designed to cover diverse, fragment-like chemical space (molecular weights 160–320 Da; hydrogen bond donors/acceptors ≤ 3), and included a range of physicochemical properties and molecular shapes (Fig. 2a, Supplementary Fig. 5 and Supplementary Table 1)[37,38]. We also included a degree of compound similarity to allow interrogation of structure-activity relationships.

The 80-member library was screened in both HEK293T and Jurkat lysates at 50 μM following incubation for 1 h. Peptide intensities measured in fragment-treated samples ($n = 4$) were compared to the DMSO control samples ($n = 16$) and reported as competition ratios (CR = Intensity$_{DMSO}$/Intensity$_{compound}$). In total, this resulted in a dataset of almost 5 million CRs and associated $p$-values, describing the interaction of the 80 fragments with the >30,000 cysteine-containing peptides detected across the two cell lines (Fig. 2b and Supplementary Fig. 6a). To focus on the most robust liganding events in each screen, we performed strict peptide filtering and have defined liganded peptides as those with statistically significant competition of at least 50% ($\log_2(CR) \geq 1.0$, $-\log_{10}(p\text{-value}) \geq 1.3$). The filters we have applied are described in detail in the experimental methods and the results on peptide numbers in Supplementary Table 2. From these 80 compounds, we detected a total of 742 unique liganding events for 438 cysteine sites from 413 proteins (Supplementary Data 1). On average, five liganded cysteines were identified per compound (Supplementary Fig. 6b). Six compounds (**PP23, PP219, PP57, PP225, PP147, PP207**) showed increased reactivity in both HEK293T and Jurkat lysate, each liganding at least 35 cysteine sites. These compounds are expected to be more promiscuous due to the presence of certain functional groups for example, electron withdrawing anilines (e.g., **PP219**) and α-substituents that disrupt the planarity of the amide bond (e.g., **PP207**).

We identified a number of ligands for proteins that already have chemical probes, however, the majority of these proteins (>80%) currently lack chemical matter to probe cellular function (i.e., Tbio or Tdark target development level) (Fig. 2c). The proteins liganded also come from a number of protein families of therapeutic value. We ligand 22 kinases; as a protein family, kinases have drawn significant interest for the development of targeted covalent inhibitors capable of disrupting cell signalling pathways (Fig. 2c)[39,40]. In addition to kinases, we liganded over 130 other enzymes, including 17 E3 ligases; these interactions could prompt the development of novel heterobifunctional molecules capable of inducing protein proximity between an E3 ligase and a protein of interest for targeted protein degradation[41,42].

Liganded cysteine sites were further classified according to their presence in protein pockets. To determine if a cysteine residue is located within a compound-accessible pocket, we applied the program Fpocket to the predicted monomer structure of the respective protein, as obtained from AlphaFold Database v4[43–45]. Across the proteome, 40% of cysteines are located in (or within 1.5 Å of) a protein pocket (Fig. 2d). When applied to our liganded sites, the percentage of cysteine residues located in pockets increases to 49%, which is further increased to 55% when considering only liganded sites of high occupancy ($\log_2(CR) \geq 2.0$). This enrichment indicates that non-covalent fragment recognition is a key contributing factor to protein engagement. However, the observation of liganding events in regions that lack detectable pockets in their structure, or in regions of protein disorder, highlights the ability of covalent ligands to target regions of proteins that have been traditionally challenging to target with non-covalent molecules.

The strength and selectivity of the strongest interactions we detected is summarised by the heatmaps in Fig. 2e (HEK293T) and Supplementary Fig. 6c (Jurkat). Several cysteine sites were strongly liganded by multiple fragments, indicating enhanced reactivity and/or high ligandability of these residues. Indeed, the four most frequently liganded cysteine sites are nucleophilic active site residues – ACAT1 Cys126, ALDH6A1 Cys317, NIT1 Cys203 and NIT2 Cys153 (liganded by 12–40% of the fragments) (Fig. 2f)[46,47]. Such interactions highlight tractable opportunities for covalent inhibition of enzyme activity. While these reactive cysteine residues are engaged by many compounds, some compounds show high selectivity for these sites, including **PP187** for ACAT1 and **PP173** for ALDH6A1 (indicated by arrows in Fig. 2e; associated volcano plots shown in Supplementary Fig. 6d) highlighting the opportunity to develop selective ligands for highly reactive residues.

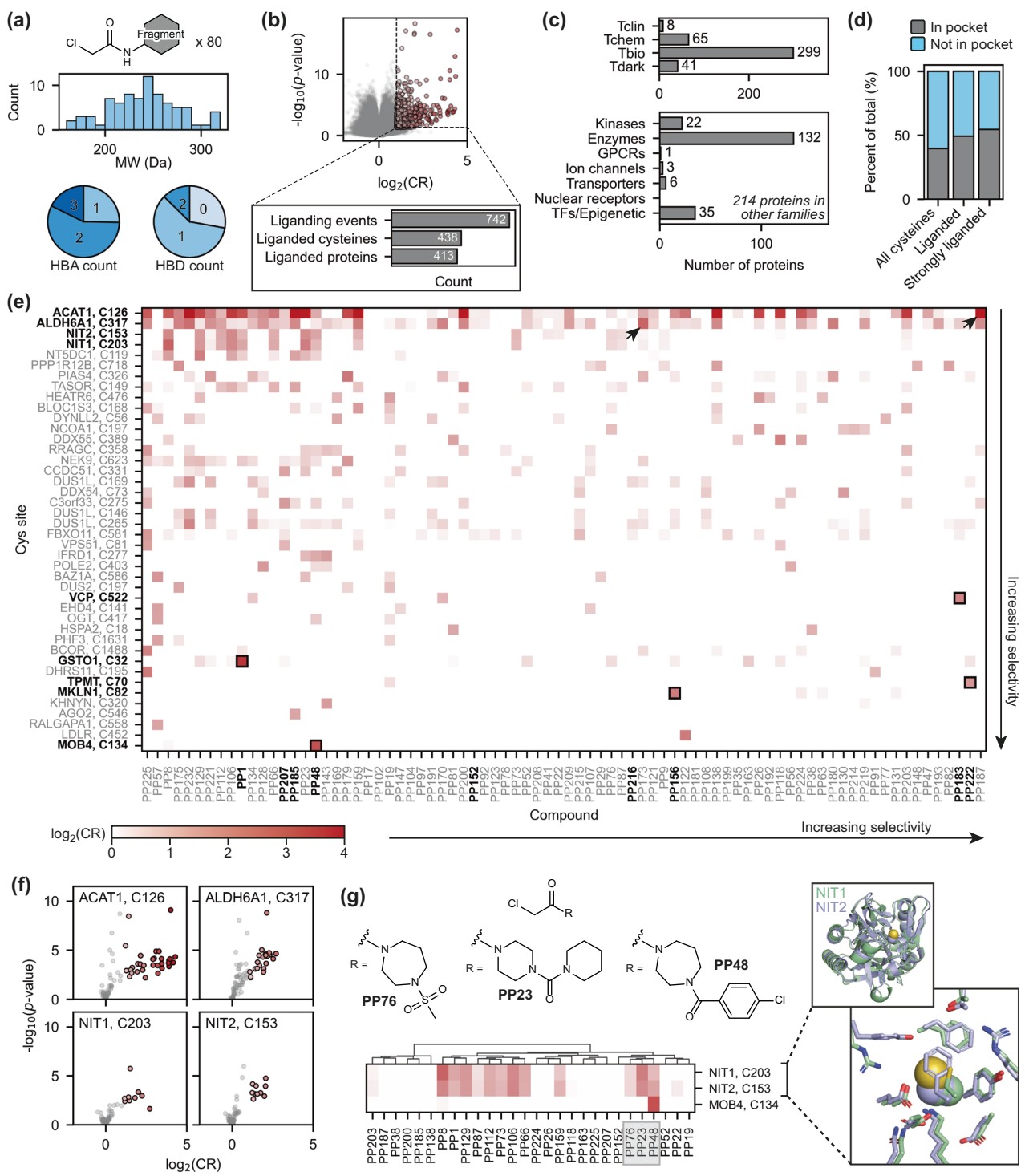

To investigate trends in the observed compound-protein pairings, hierarchical clustering was performed, grouping compounds based on their molecular fingerprints (Morgan) and proteins based on their competition ratios across the library[48]. This clustering approach highlighted similar binding profiles between structurally similar compounds and proteins. Notably, NIT1 and NIT2 proteins showed near-identical binding profiles to the fragment library, in particular to hindered tertiary chloroacetamides, consistent with the homology of the active sites of these proteins (Fig. 2g). The selectivity of this enzyme family for these compounds demonstrates the potential to identify molecular chemotypes that can be developed towards activity-based protein probes[47].

Of particular interest were instances where specific interactions were observed between non-hyperreactive cysteines and non-promiscuous fragments[49]. We determined the selectivity of each detected interaction by calculating the difference between the CR for a given interaction and the mean of the five strongest interactions that were detected both for that cysteine site and compound, respectively. By this metric, the top five most selective interactions that were detected in both HEK293T and Jurkat lysate were: MOB4 (Cys134) with **PP48**, MKLN1 (Cys82) with **PP156**, VCP (Cys522) with **PP183**, TPMT (Cys70) with **PP222**, and the active site residue of GSTO1 (Cys32) with **PP1** (highlighted by boxes in the heatmaps in Fig. 2e; volcano plots shown in Fig. 3)[50]. The interaction between GSTO1 Cys32 and **PP1** was

**Fig. 2 | The 80-compound screen performed in HEK293T and Jurkat lysate using HT-LFQ chemoproteomics. a** Distribution of molecular weights (MW) and hydrogen-bond acceptor/donor (HBA/HBD) counts for the 80 chloroacetamide fragments in the screening library. **b** Volcano plot showing an overlay of all the data obtained in this experiment, where each data point represents the interaction measured in each cell line between a fragment (50 µM) and a cysteine-containing peptide. Liganding events are defined where a fragment shows strong, statistically-significant competition (measured by a competition ratio, CR) with IA-DTB: $\log_2(CR) \geq 1$ and $-\log_{10}(p\text{-value}) \geq 1.3$, as indicated by dotted lines. This experiment was performed with technical replicates ($n = 4$ for compound-treated samples, $n = 16$ for DMSO control samples) in both cell lines. **c** Distribution of liganded proteins across 'Illuminating the Druggable Genome' protein families (top) and target development levels (bottom). GPCRs: G protein-coupled receptors; TFs: transcription factors. **d** The proportion of cysteine residues that lie within or near

pockets (grey) increases when considering cysteines that are liganded ($\log_2(CR) \geq 1$) or strongly liganded ($\log_2(CR) \geq 2$) by at least one fragment, compared to the cysteinome as a whole. **e** Heatmap of all the interactions detected in HEK293T lysate. For clarity, this heatmap only includes cysteines that are liganded by at least one compound with $\log_2(CR) \geq 1.5$. **f** Volcano plots showing interactions detected in HEK293T lysate with active site cysteine residues – these residues show binding to a high proportion (≥10%) of fragments in the screening library. **g** NIT1 and NIT2 show very similar binding profiles to tertiary chloroacetamides (heatmap is clustered by molecular fingerprint), which reflects the high structural similarity of these proteins. The colour scheme used for the volcano plots/heatmap in (**f, g**) matches that used in (**e**). In the NIT1 and NIT2 structures (AlphaFold2)[44], the side chain of the liganded cysteine residue (Cys203 and Cys153, respectively) is shown as spheres. All $p$-values were calculated using Welch's $t$-test (two-sided). Source data are provided as a Source Data file.

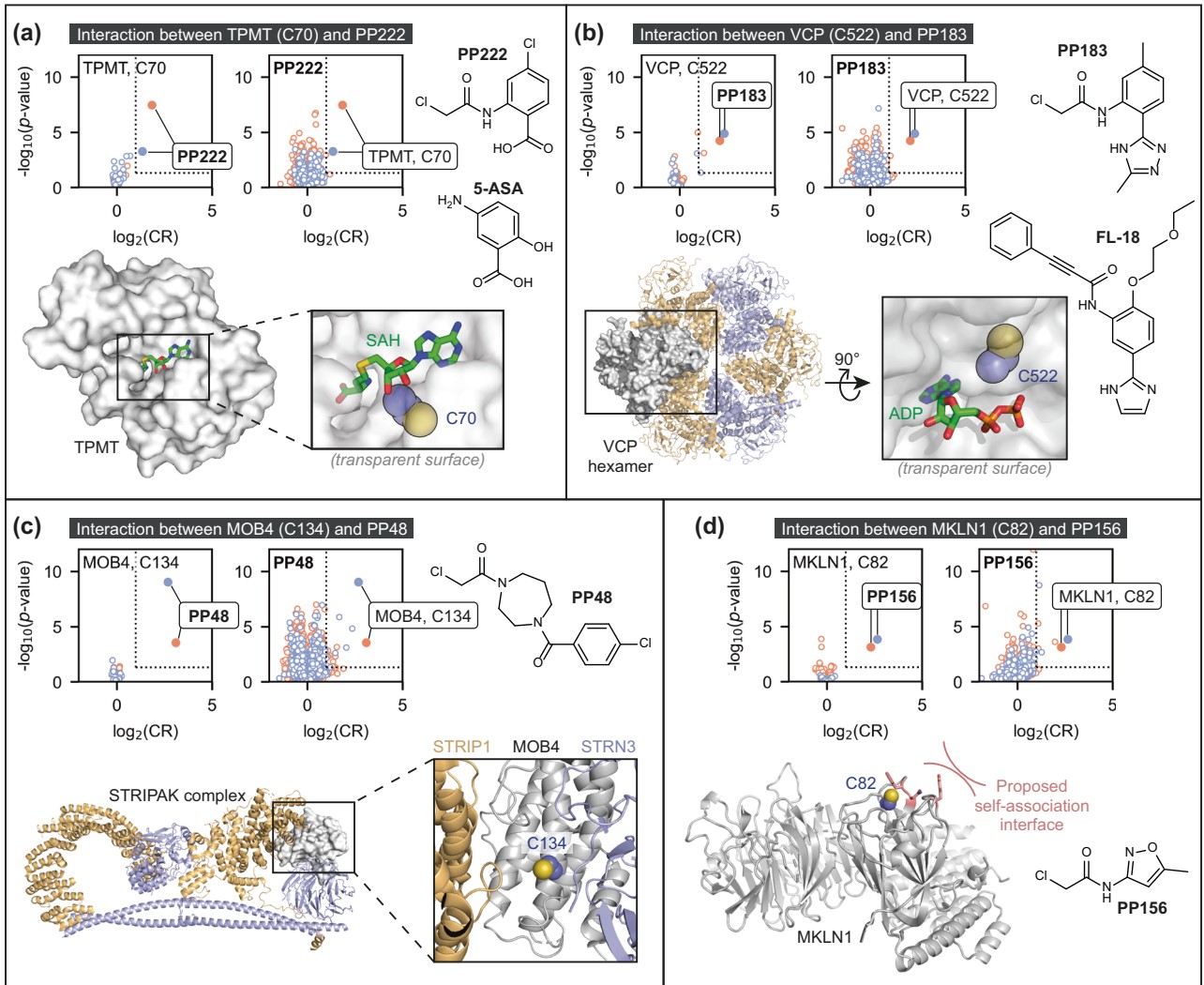

**Fig. 3 | Top four specific protein-fragment interactions detected in the initial screen.** Specific interactions were detected with TPMT Cys70 (**a**), VCP Cys522 (**b**), MOB4 Cys134 (**c**), and MKLN1 Cys82 (**d**), based on comparison of compound-treated samples (50 µM, $n = 4$) with DMSO control samples ($n = 16$). Volcano plots are shown both for the cysteine residue (left) and compound (right) involved in each interaction, to highlight that each of the four cysteine residues is only liganded by one fragment in both HEK293T (orange) and Jurkat (blue) lysate, and that this cysteine site is most strongly competed target of these compounds. Dotted lines indicate the thresholds

used to identify liganding events: $\log_2(CR) \geq 1$ and $-\log_{10}(p\text{-value}) \geq 1.3$. All $p$-values were calculated using Welch's $t$-test (two-sided). Protein structures are either from the Protein Data Bank (PDB) or AlphaFold2: TPMT, PDB ID 2H11 (residues 40–245)[74]; VCP, PDB ID 5FTJ[75]; STRIPAK (containing MOB4), PDB ID 7K36[76]; MKLN1, AlphaFold2 model[44]. The side chain of each targeted cysteine residue is shown as blue and yellow spheres (for the carbon and sulfur atoms, respectively). SAH S-adenosyl-L-homocysteine, ADP adenosine diphosphate. Source data are provided as a Source Data file.

deprioritised for follow up, as it has been liganded by covalent fragments in multiple other datasets[50].

Interestingly, previously reported TPMT and VCP inhibitors bear strong resemblance to the ligands we have identified. Thiopurine

S-methyltransferase (TPMT) activity is inhibited by a range of non-covalent benzoic acid derivatives (*e.g.*, 5-aminosalicylic acid, 5-ASA), and our hit fragment, **PP222**, contains the same benzoic acid core (Fig. 3a)[51]. The targeted cysteine, Cys70, lies within a buried substrate-

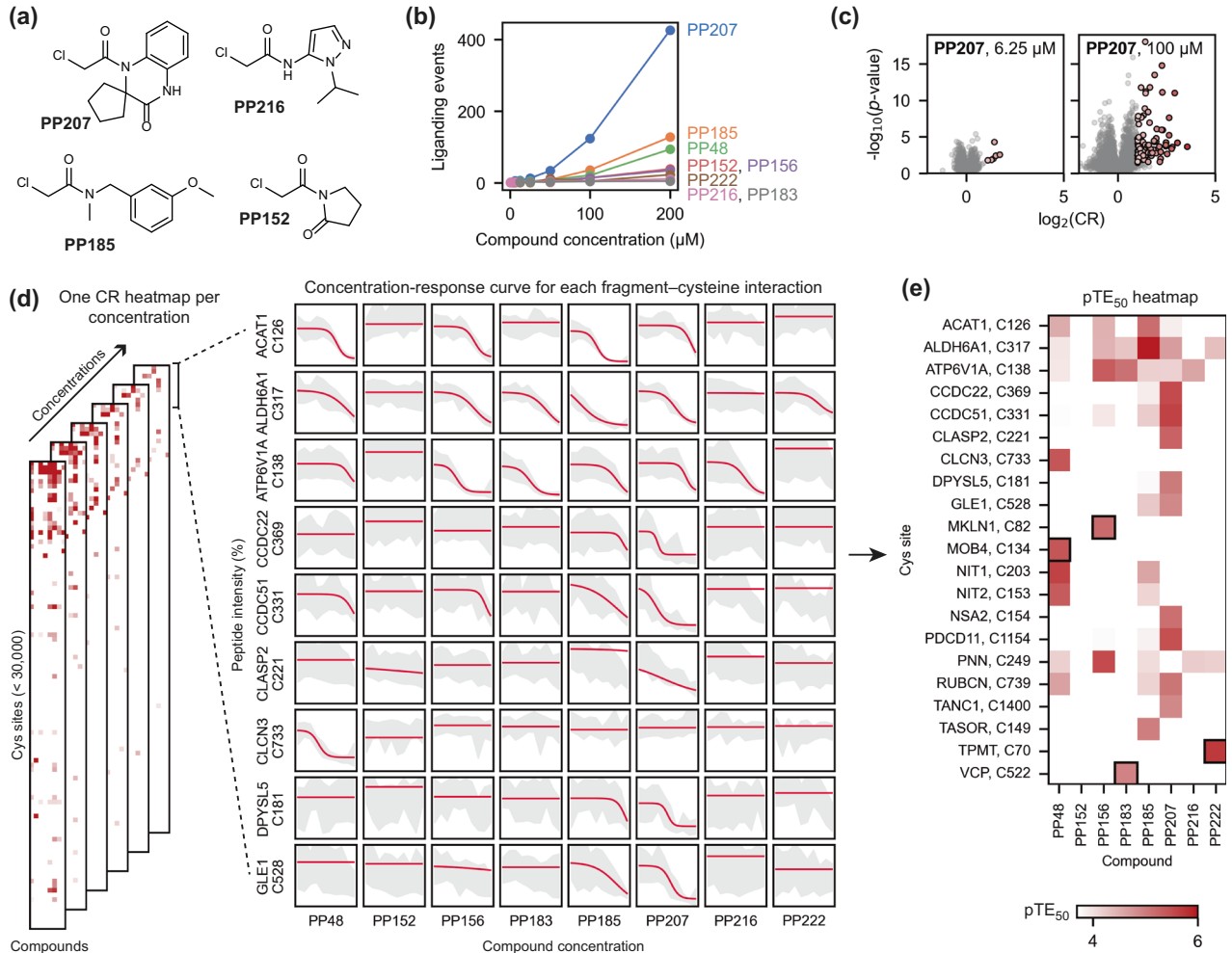

**Fig. 4 | Concentration-response chemoproteomics experiment. a** The structures of four compounds that showed a range of promiscuity levels in the initial screen. These four compounds, along with the four compounds that showed specific interactions with a protein target (Fig. 3), were tested in a 10-point concentration-response experiment in HEK293T lysate ($n$ = 4 for compound-treated samples; $n$ = 25 DMSO control samples). The total number of liganding events detected for each of these eight compounds varied widely across the concentration range tested (**b**, **c**); $p$-values were calculated using Welch's $t$-test (two-sided). **d** The concentration-response experiment was analysed by performing logistic regression to identify any concentration-dependent interactions between each compound and all detectable cysteine residues. **e** Heatmap showing the $pTE_{50}$ values of all concentration-dependent interactions that were confidently identified in this experiment. The selective interactions that were identified in the initial screen are highlighted by black boxes. Source data are provided as a Source Data file.

binding cavity of this enzyme, with a high density of basic residues nearby that can engage acidic small molecules. VCP is a homohexameric ATPase that has been targeted for the treatment of multiple diseases including acute myeloid leukemia. Covalent inhibitors have been identified that target Cys522, which lies in one of the nucleotide binding pockets, to inhibit enzyme activity and cell growth[52-54]. One of these inhibitors, FL-18, bears structural similarity to the 5-6-fused heterocyclic ring system of **PP183** (Fig. 3b).

Both MOB4 and MKLN1 belong to the Tbio target development category and therefore have no known chemical tools to probe cellular function. The globular adaptor protein, MOB4, is a key component of STRIPAK (striatin-interacting phosphatase and kinase) complexes that play key roles in regulating diverse cellular functions, including cell cycle control and motility[55,56]. MOB4 performs a scaffolding function in STRIPAK complexes, and Cys134 lies adjacent to two protein-protein interaction interfaces (Fig. 3c). Therefore, **PP48** offers a route to a tool molecule to modulate complex formation or dissociation, and thus STRIPAK function[57,58]. MKLN1 (muskelin) is part of the CTLH complex, a multi-subunit RING E3 ubiquitin ligase[59]. **PP156** binds to Cys82 which lies adjacent to a proposed self-association interface in muskelin (Fig. 3d) and could be used to probe the functional relevance of this

interface[60]. Intriguingly, neither the MOB4 or MKLN1 cysteine sites are located in pockets according to Fpocket analysis of monomeric protein structures, and thus these ligands may bind to pockets formed in multimeric protein complexes.

## Proteome-wide concentration-response analysis

The throughput of the HT-LFQ chemoproteomics platform enables fragment screening at multiple concentrations to accurately assess the potency and selectivity of liganding events. We selected eight compounds for screening in HEK293T cell lysate via 10-point concentration-response (0.4–200 µM, quadruplicate measurements). Four of these compounds were those that were identified to form highly selective interactions in the initial screen: **PP183** (VCP), **PP222** (TPMT), **PP156** (MKLN1), and **PP48** (MOB4). This compound set was supplemented with four additional fragments that varied in their overall promiscuity: **PP207** (high promiscuity), **PP156** (medium promiscuity), **PP152** (low promiscuity), and **PP216** (low promiscuity) (Fig. 4a).

From this experiment, we identified 761 liganding events at the highest concentration of 200 µM, tenfold greater than the number of interactions detected at 50 µM (81 liganding events) (Fig. 4b). The overall promiscuity of the compounds screened broadly matched the

initial screen, with a significant range in the number of cysteine sites engaged by different compounds. The high promiscuity of compounds such as **PP207** (which bound 426 cysteine sites at 200 μM) highlight its utility as a 'scout-like' fragment to identify cysteine residues across the proteome amenable to covalent modification[61]. For the lowly reactive compounds **PP152** (39 sites) and **PP216** (11 sites), screening at a high concentration can be used to identify selective interactions that can be optimised. These data also illustrate how the screening concentration used can affect the apparent selectivity of a compound (*e.g.*, **PP207** at 100 μM and 6.25 μM) (Fig. 4c), highlighting the value of methods that offer sufficient throughput and flexibility in experimental design to allow screening at multiple concentrations.

Screening at multiple concentrations allows for curve fitting to obtain half-maximal target engagement ($TE_{50}$) values for each fragment-cysteine interaction. After filtering, based on potency and the quality of curve fitting, a heatmap was generated to enable visualisation of concentration-dependent binding events, which showed strong agreement with the interactions detected in the initial single-shot screen (Fig. 4d, e and Supplementary Fig. 7). The four compound-protein pairs of interest were confirmed with the following $pTE_{50}$ (*i.e.*, $-\log_{10}(TE_{50})$) values: MOB4-**PP48** $pTE_{50} = 5.4 \pm 0.1$, VCP-**PP183** $pTE_{50} = 4.9 \pm 0.1$, TPMT-**PP222** $pTE_{50} = 5.7 \pm 0.4$, and MKLN1-**PP156** $pTE_{50} = 5.2 \pm 0.2$. None of these four cysteine sites were liganded by any other fragment at any concentration, highlighting the specificity of the cysteine site for the respective fragment (Fig. 5a), and no other peptides from these proteins showed a concentration-dependent change in intensity upon fragment treatment (Supplementary Fig. 8). Each compound had no more than three off-targets within $\Delta pTE_{50} \leq 0.5$, including sites commonly bound by chloroacetamide fragments, such as ALDH6A1 Cys317 and ATP6V1A Cys138 (Fig. 5b)[50].

### HT-LFQ chemoproteomics for hit expansion and live cell treatments

**PP48**, which bound to MOB4 Cys134, as well as to the active site cysteines NIT1 Cys203 and NIT2 Cys153, was selected for hit expansion to explore how our chemoproteomics platform could be used to drive structure-activity relationships (SAR). The non-covalent core of **PP48** contains three molecular features: a diazepane ring, an amide linker, and a substituted aromatic ring. Binding profiles from structurally-similar compounds that were tested in the initial library screen suggested an important role for the terminal aromatic ring in MOB4 binding (Fig. 2g). To further explore SAR around **PP48**, we designed seven additional analogues (Supplementary Fig. 9a), varying the nature of the linker, the substituents on the aromatic system, and the structure of the diazepane ring. Each analogue (**PP48a-g**) was screened in five-point concentration-response (3–50 μM) in HEK293T lysate.

This screen highlighted divergent SAR and opportunities to drive compound selectivity. We identified several useful control compounds: **PP48a, PP48d** and **PP48e** (Fig. 5c and Supplementary Fig. 9b). Compared to **PP48, PP48a** (chloro to methoxy) and **PP48e** (amide to urea linker and loss of aromatic substituent) both showed similar engagement to the primary off-targets NIT1 and NIT2, but showed either reduced or completely abolished binding to MOB4. Conversely, for **PP48d**, where the amide carbonyl is removed, engagement of NIT1 and NIT2 is significantly reduced compared to MOB4. These compounds could therefore act as useful controls in deconvoluting the effects of on-target and off-target binding in functional assays.

The aforementioned screening and concentration-response experiments were all performed in cell lysates to maximise sample throughput. However, any hits identified through such screens have most applicability in functional and phenotypic experiments performed in live cells. Therefore, live HEK293T cells were treated with **PP48, PP48a, PP48d** and **PP48e** at 25 μM for 2 h, followed by cell lysis, IA-DTB treatment, and quantification of modified peptides (Fig. 5c). Importantly, we confirmed the binding of **PP48** to MOB4 Cys134, NIT1 Cys203 and

NIT2 Cys153, with additional off-targets also identified (e.g., SORD Cys45), potentially due to the increased incubation time employed for the live cell treatment (1 h vs 2 h) or increased temperature (room temperature vs 37 °C) (Supplementary Fig. 9c). The selectivity profile of the control compounds, **PP48a, PP48d** and **PP48e**, also showed concordance between lysate and cell treatment. These results highlight that profiling of compounds in lysates is an effective method to improve throughput and simplify workflows while being an effective surrogate for measurement of interactions formed in live cells[62].

## Discussion

Identifying chemical tools for the unliganded proteome is essential to accelerate the exploration of protein function and the discovery of therapeutic opportunities. Various initiatives, such as 'Target 2035', have been established with the aim to identify pharmacological modulators for every protein in the human proteome[7,35]. Achieving this ambition requires the development of high-throughput platforms to screen molecules in complex cellular environments and identify novel protein-ligand interactions. Platforms that combine competitive profiling of covalent fragments with chemoproteomics have drawn interest from academia and industry, allowing screens to be performed in cells and lysates, and providing a quantitative readout of fragment-protein engagement.

We have developed a label-free chemoproteomics platform for screening reactive fragments across the proteome by competition with a hyper-reactive IA-DTB probe. We have moved away from recently reported DDA and TMT labelling-based platforms to overcome limitations of these methods associated with incomplete datasets and batch effects. By employing LFQ and DIA, our platform offers high analytical throughput, deep cysteinome coverage, and high sample reproducibility and data completeness, whilst avoiding the need for costly isotopic labelling reagents.

Our LFQ based chemoproteomics platform allows for comparison of peptide quantities between an unlimited number of samples. This offers excellent flexibility in experimental design, facilitating screening of large libraries at multiple concentrations. Performing DIA ensures every peptide in the sample is fragmented regardless of abundance, maintaining data reproducibility and completeness. Furthermore, an extra level of peptide separation was performed via trapped ion-mobility using PASEF (on a Bruker timsTOF Pro 2), which provides unrivalled cysteinome depth for DIA-based cysteine profiling (~23,000 cysteines identified per run; ~30,000 per experiment)[29]. Finally, dia-PASEF enabled short (21-min) chromatographic gradients without compromising identification depth[31]. A remaining limitation in chemoproteomics studies is the overall cysteinome coverage. Our HT-LFQ chemoproteomics platform achieves coverage of ~40% of the proteome but less than 15% of the entire cysteinome. From our analysis, protein abundance, lysis conditions and peptide properties were key factors in determining cysteine detection, and so we anticipate that performing screens in different cell lines with varied protein expression profiles maybe beneficial in improving cysteine coverage.

We evaluated the platform by profiling a reactive fragment library of 80 chloroacetamides in quadruplicate in two cell lines, producing a robust dataset of liganding interactions across the cysteinome. The screen identified 438 liganded cysteine sites from 413 proteins, including over 300 proteins from the Tbio and Tdark target development categories, for which no chemical tools exist[3,7,35]. While examination of the location of liganded cysteines highlighted an enrichment of cysteines near pockets, many liganded cysteines were in regions predicted to be disordered and many others may lie adjacent to pockets that only develop upon the formation of protein complexes[63]. Together, these observations highlight the value of screening compounds against proteins in an endogenous setting, and the potential for covalent compounds to ligand proteins previously considered to be undruggable. The majority of the interactions displayed good

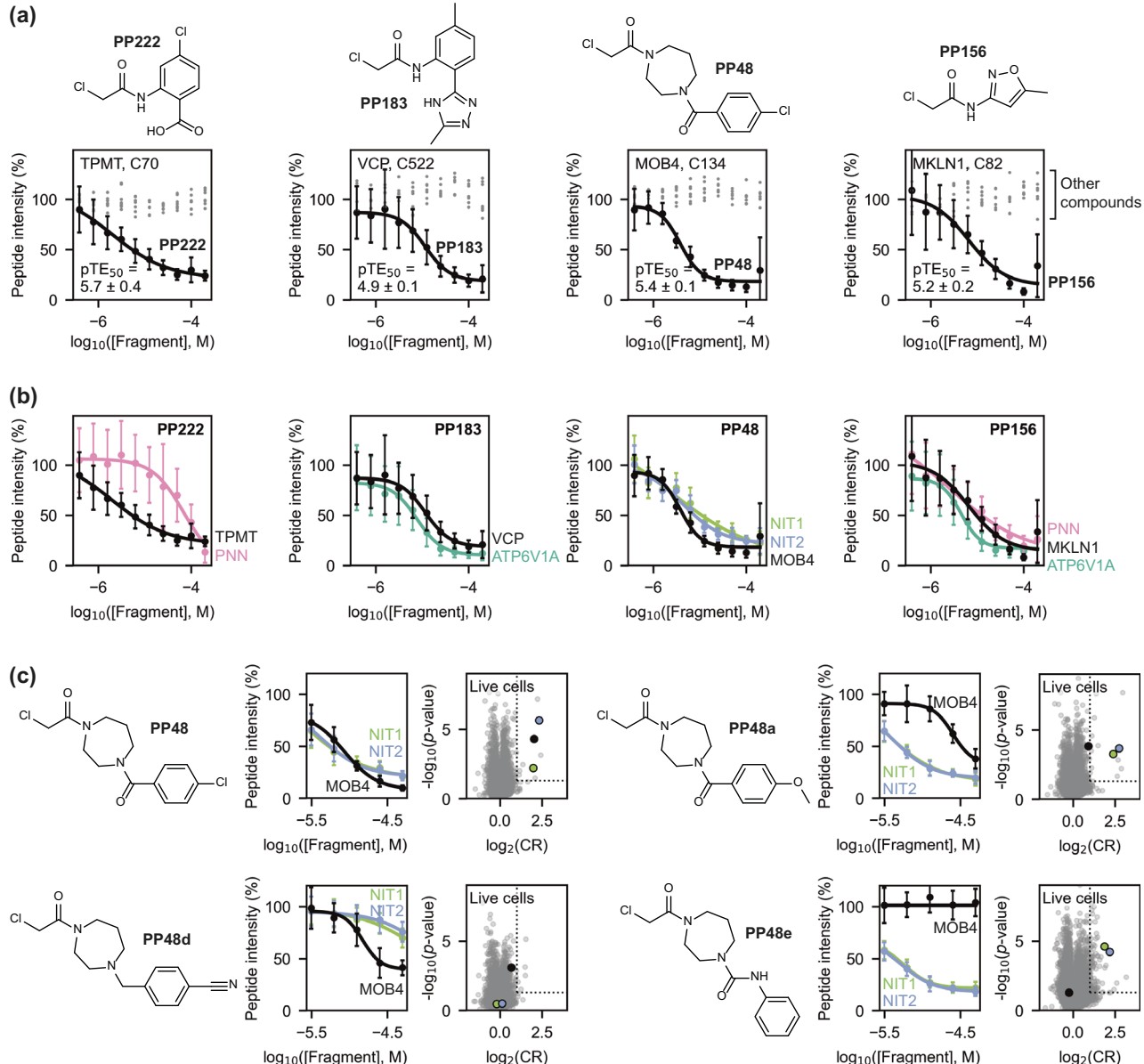

**Fig. 5 | The selectivity and potency of prioritised protein-fragment interactions. a** Concentration-response data (mean ± standard deviation) obtained for VCP Cys522, TPMT Cys70, MOB4 Cys134 and MKLN1 Cys82 in HEK293T lysate ($n = 4$ for compound-treated samples; $n = 25$ for DMSO control samples), highlighting the concentration-dependent engagement of these cysteines with their identified ligand (PP222, PP183, PP48 and PP156, respectively; black data), in contrast with the data obtained for the other seven compounds tested in the experiment (grey data points; only mean values are shown for clarity). **b** The strongest concentration-dependent interactions detected for PP222, PP183, PP48 and PP156, showing other cysteine residues engaged by these compounds in addition to those shown in (**a**). These off-target cysteine residues are as follows: PNN Cys249 (PP222 pTE$_{50}$ = 4.2 ± 0.4; PP156 pTE$_{50}$ = 5.5 ± 0.7), ATP6V1A Cys138 (PP183 pTE$_{50}$ = 5.1 ± 0.1;

PP156 pTE$_{50}$ = 5.3 ± 0.1), NIT1 Cys203 (PP48 pTE$_{50}$ = 5.6 ± 0.8), and NIT2 Cys153 (PP48 pTE$_{50}$ = 5.4 ± 0.1). **c** SAR analysis around the interaction between PP48 and MOB4 Cys134, NIT1 Cys203 and NIT2 Cys153, showing the results of five-point concentration-response data acquired in HEK293T lysate (1-h incubation; $n = 4$ for compound-treated samples; $n = 25$ for DMSO control samples) and data acquired in live HEK293T cells (25 μM, 2-h incubation; $n = 3$ for compound-treated samples; $n = 12$ for DMSO control samples). Precise pTE$_{50}$ values for these interactions and for other compounds in the SAR compound set can be found in Supplementary Fig. 9. Unless otherwise stated, concentration-response data is shown as mean ± standard deviation (data points ± error bars) alongside the logistic regression curve. All *p*-values were calculated using Welch's *t*-test (two-sided). Source data are provided as a Source Data file.

selectivity (with each compound engaging only ~5 cysteines, on average, at 50 μM), representing promising starting points for the development of chemical probes. Furthermore, screening of hit fragments across a concentration gradient produced rich datasets of concentration-response curves for every detected cysteine residue, allowing for prioritisation of compounds for further development based on potency and proteome selectivity.

The throughput and flexibility of the platform facilitates the exploration of structure-activity relationships by chemoproteomics.

We explored the selectivity and potency of structural analogues of **PP48**, which engaged the adaptor protein MOB4 as well as cysteine residues in NIT1 and NIT2. SAR analogues showed differential binding profiles, highlighting opportunities to improve selectivity for MOB4 or NIT1/2. While the majority of screening was performed in cell lysates to facilitate plate-based workflows, a number of SAR compounds were profiled in live cells, validating the interactions detected in lysate-based experiments. The simultaneous quantification of cellular on-target potency as well as proteome-wide off-target binding is a key

benefit for chemoproteomics screening of covalent compounds and has the potential to streamline early drug discovery efforts. This information is particularly challenging to obtain when screening non-covalent compounds in cellular assays or screening against purified proteins.

Looking forward, we anticipate application of this robust screening platform to profile larger compound libraries against the native proteome. The resulting full matrix datasets will offer opportunities for machine learning approaches to predict ligandability and drive iterative library design towards selective chemical probes, and thus expand the liganded proteome.

# Methods
## Compounds
All chloroacetamide fragments were purchased from Enamine (catalogue numbers provided in Supplementary Table 1) and stored either as solids at −20 °C or as DMSO stocks at −80 °C. Iodoacetamide desthiobiotin (IA-DTB) was synthesised according to literature reports[18]. The product was purified via silica gel chromatography (ISCO 80 g RediSep Gold column, 0–20% methanol in dichloromethane, 60 ml/min flow rate, dry loaded as silica gel powder). Fresh stock was prepared from solid at a concentration of 50 mM in DMSO and used immediately.

## Cell culture
HEK293T cells were maintained at 37 °C, 5% $CO_2$ in Dulbecco's modified eagle medium (Gibco, 41966-029) supplemented with 10% (v/v) fetal bovine serum (Gibco, 10270-106) and 1% v/v penicillin–streptomycin-glutamine (Gibco, 10-378-016). Jurkat cells were maintained at 37 °C, 5% $CO_2$ in RPMI-1640 GlutaMAX (Sigma Aldrich, R8758) supplemented with 10% v/v fetal bovine serum (Gibco, 10270-106) and 1% v/v penicillin-streptomycin (Sigma Aldrich, P4333).

## Preparation of chemoproteomics samples
**Lysate compound treatment.** Cell pellets (pre-washed twice with PBS) were suspended in RIPA lysis buffer (150 mM NaCl, 1.0% IGEPAL CA-630 (Sigma Aldrich, I8896), 0.5% sodium deoxycholate (Sigma Aldrich, D6750), 0.1% SDS (Fisher Scientific, 10607633), 50 mM HEPES, pH 8.0) containing protease inhibitor cocktail (Sigma Aldrich, P1860). Samples were sonicated with a Branson probe sonicator (3–5 × 2 s pulses, 10% amplitude) and filtered through a 0.22 μM filter. The protein concentration of the lysate was determined using a Pierce™ BCA assay kit (Thermo Scientific, 23227) according to manufacturer's instructions. Lysate was diluted to the desired concentration and used on the day of lysis.

For cell lysate treatment, compounds were first prepared at an appropriate concentration in DMSO (2 μL) in a 96-well plate. To this, cell lysate (200 or 500 μg protein) was added to each well to reach a final volume of 200 μL and the plates were then shaken on a Thermomixer (room temperature (rt), 600 rpm, 1 h). Experiments were performed with technical replicates, where multiple identical samples were prepared in parallel.

**Live cell compound treatment.** HEK293T cells were cultured in 10 cm dishes to 90% confluency. Each plate was treated with a compound (25 μM, final DMSO concentration of 0.4% v/v) or DMSO alone for 2 h at 37 °C. Media was removed and cells were washed three times with PBS. Cell lysis and protein quantification methods were performed as reported above, with samples clarified by centrifugation (16,000 rcf, 10 min, 4 °C) instead of filtration. Replicate samples were prepared from cells that had been grown separately for one doubling time. Compound treatment and subsequent sample preparation on these replicate samples was performed in parallel.

**IA-DTB treatment.** IA-DTB (500 μM; fresh DMSO stock) was added to wells containing compound-treated cell lysates on 96-well plates (rt,

600 rpm, 1 h; performed in the dark). Following treatment, samples were reduced with dithiothreitol (Thermo Scientific, R0861) (5 mM, rt, 600 rpm, 30 min) and alkylated with iodoacetamide (Thermo Scientific, 122270250) (10 mM, rt, 600 rpm, 30 min; performed in the dark).

**Glass bead slurry preparation.** To prepare stock glass bead mixtures, glass spheres (Supelco, 440345) were first suspended at a concentration of 100 mg/mL in ultrapure water (LC/MS grade)[30]. The resulting slurry was then vortexed and centrifuged (1500 rcf, 5 min, 4 °C), and the buoyant beads were gently aspirated to leave a glass bead pellet. This process was repeated twice with ultrapure water and twice with acetonitrile. After the final wash, the bead pellet was resuspended in acetonitrile (making the solution up the same volume as the original 100 mg/mL solution in water), and a final concentration of 50 mg/mL was assumed, as half of the beads are typically lost during the washing procedure. The beads were stored at 4 °C until needed.

**Glass bead assisted sample clean up and digestion.** Glass beads from the pre-prepared stock solution were diluted in acetonitrile (6.25 mg/mL) and this bead slurry was dispensed into MultiScreen deep filter plates (Merck Millipore, MDRLN0410) (800 μL/well). IA-DTB-treated cell lysates were transferred to the bead slurry and agitated (600 rpm, 5 min), inducing protein precipitation. The plates were centrifuged (1500 rcf, 2 min) to remove the supernatant and washed with 80% v/v ethanol (1500 rcf, 2 min, 3×). Proteins were resolubilised in HEPES (50 mM, pH 8.5, 250 μL/well) containing Pierce™ Trypsin Protease (Thermo Scientific, 90059) (1:100 enzyme/protein ratio) and digested overnight (rt, 800 rpm). Peptides were recovered into collection plates through centrifugation (1500 rcf, 5 min) and subsequent washes of the glass beads with HEPES (50 mM, pH 8.5, 2 × 75 μL/well).

**Enrichment.** Peptide solutions were transferred to a sealed Microlute plate (Porvair Sciences, 240002). Pierce™ High Capacity NeutrAvidin™ Agarose resin (Thermo Scientific, 29204) was prepared by washing with HEPES (50 mM, pH 8.5, 3×), and then dispensed (50 μL of slurry) into each well of the sealed Microlute plate and agitated with the peptides (800 rpm, 2 h). The drain cap seal was removed and the plate was centrifuged (700 rcf, 1 min) to remove the supernatant. Beads in each well were washed with 0.1% SDS in HEPES (50 mM, pH 8.5; 850 μL/well, 3×), HEPES (50 mM, pH 8.5, 850 μL/well, 3×), and finally ultrapure water (850 μL/well, 3×). Enriched peptides were eluted from the neutravidin resin with 1:1 acetonitrile/water, containing 0.1% formic acid (200 μL/well, 700 rcf, 1 min). This elution step was repeated (100 μL/well, 2×). The collection plate was frozen and samples were dried using a Speedvac at 4 °C. Plates were stored at −80 °C.

**Desalting.** C18 Nest desalting plates (The Nest Group Inc, HNS S18V) were conditioned with acetonitrile (300 μL/well) and centrifuged (50 rcf, 1 min). Plates were equilibrated with ultrapure water/0.1% trifluoroacetic acid (300 μL/well, 2×) and centrifuged (500 rcf, 5 min). Samples were re-dissolved in ultrapure water/0.1% trifluoroacetic acid (200 μL/well) on a Thermomixer (rt, 600 rpm, 5 min), then loaded into the prepared C18 NEST plate(s) and centrifuged (500 rcf, 5 min). Samples were washed with ultrapure water/0.1% trifluoroacetic acid (200 μL/well, 2×). Peptides were eluted using 1:1 acetonitrile/water with 0.1% trifluroacetic acid (150 μL/well, 2×) by centrifugation (500 rcf, 5 min). The collection plate was frozen and samples were dried using a Speedvac at 4 °C. Plates were stored at −80 °C.

## Preparation of global proteomics samples
**Sample preparation and peptide recovery.** HEK293T cell pellets were lysed in RIPA lysis buffer according to the lysis procedure reported above for chemoproteomics samples. Lysate was prepared at 1 μg/μL and diluted further to 0.5 μg/μL with S-Trap lysis buffer (10% SDS (Fisher Scientific, 10607633), 100 mM triethylammonium bicarbonate

(Sigma Aldrich, T7408), TEAB, pH 7.55). Samples were reduced and alkylated as reported above. Samples were then acidified with 10 μL of 12% phosphoric acid (Supelco, PX1000). S-Trap binding buffer (750 μL, 90% methanol, 100 mM, TEAB, pH 7.1) was added to each sample and gently agitated. Acidified mixtures were transferred into wells of a 96-well S-Trap™ plate (Protifi, C02-96well-1) and centrifuged (1500 rcf, 2 min). Captured protein was washed with S-Trap binding buffer (1500 rcf, 2 min, 3×) and digested with S-Trap digestion buffer (125 μL, 50 mM TEAB) containing Pierce™ Trypsin Protease (1:25 enzyme/protein ratio) (47 °C, 1 h). Following digestion, S-Trap digestion buffer (80 μL) was added and peptides were collected into a 96 well plate by centrifugation (1500 rcf, 2 min). Each well was washed with 80 μL of 0.1% aqueous formic acid, followed by 80 μL of 50% aqueous acetonitrile containing 0.1% formic acid. The collection plate was frozen and samples were dried using a Speedvac at 4 °C. Plates were stored at −80 °C.

## Liquid chromatography-mass spectrometry

**High pH off-line fractionation.** To generate a library of IA-DTB modified peptides, high pH off-line fractionation was performed. High pH off-line fractionation was either performed with an XBridge BEH C18 XP column (2.5 μm × 3 mm × 130 mm, Waters, 186006710) coupled to an UltiMate 3000 HPLC system, or with a Pierce™ High pH Reversed-Phase Peptide Fractionation Kit (Thermo Scientific, 84868). For the off-line fractionation on the XBridge BEH C18 column a 60-min acetonitrile gradient (1–35% acetonitrile) was performed at a flow rate of 200 μL/min using the following buffers: 10 mM ammonium hydroxide pH 10 (buffer A), and 90% acetonitrile, 10% 100 mM ammonium hydroxide pH 10 (buffer B). 24 fractions were consolidated and dried using a Speedvac at 4 °C. For the off-line fractionation kit, samples were prepared according to manufacturer's instructions. These samples were dried using a Speedvac at 4 °C.

**Evotip sample loading.** Prepared samples were re-dissolved in Optima™ water/0.1% formic acid (100 μL/well) on a Thermomixer (rt, 600 rpm, 5 min). Evotips (Evosep, EV2001 or EV2011) were conditioned according to manufacturer's instructions. Approximately 200 ng (the desired loading) of each peptide mixture spiked with indexed retention time (iRT) peptides (Biognosys, Ki-3002-1) was loaded onto a conditioned Evotip and queued on an Evosep One liquid chromatography system with the pre-defined 30 and 60 SPD (samples per day) methods and the corresponding Evosep Performance columns (Evosep, EV1109 and EV1137). Mobile phases A and B were 0.1% (v/v) formic acid in water and 0.1% (v/v) formic acid in acetonitrile, respectively.

**Mass spectrometry: general.** The Evosep One was coupled online to a hybrid (trapped ion mobility spectrometry) TIMS quadrupole TOF (time of flight) mass spectrometer (Bruker timsTOF Pro 2) via a captive spray nano-electrospray ion source. All chemoproteomics samples were analysed with an ion mobility range from 1/K0 = 1.638 to 0.6 Vs cm$^{-2}$ and global proteomics samples were analysed with an ion mobility range from 1/K0 = 1.6 to 0.6 Vs cm$^{-2}$. Equal ion accumulation time and ramp times were applied in the dual TIMS analyser of 100 ms each. Mass spectra were recorded from 100–1700 m/z. The ion mobility dimension was calibrated regularly using all three ions from an Agilent electrospray ionisation LC/MS tuning mix (m/z, 1/K0: 622.0289, 0.9848 Vs cm$^{-2}$; 922.0097, 1.1895 Vs cm$^{-2}$; and 1221.9906, 1.3820 Vs cm$^{-2}$). When operating the mass spectrometer in ddaPASEF mode, 10 PASEF/MS-MS scans were used per topN acquisition cycle. Singly charged precursors were excluded by their position in the m/z-ion mobility plane, and precursors that reached a target value of 20,000 arbitrary units were dynamically excluded for 0.4 min. When operating the mass spectrometer in diaPASEF mode, 8 diaPASEF scans per TIMS-MS scan were used, giving a duty cycle of 0.96 seconds. For chemoproteomics DIA analysis, variable ion mobility windows were used with fixed mass windows of 25 m/z and with

a mass range of 400–1000 m/z (Supplementary Table 3). For global proteomics DIA analysis, recent developments in window optimisation were incorporated to allow for variable ion mobility windows and variable mass windows, between a mass range of 262.18–1398.68 m/z (Supplementary Table 4)[64].

**Chemoproteomics data analysis.** Mass spectrometry raw files for chemoproteomics were analysed using Spectronaut (Biognosys; version 16).

**Generation of a hybrid library of IA-DTB modified peptides.** In total, 72 mass spectrometry files were used to generate a hybrid DDA/DIA library of IA-DTB modified peptides from both HEK293T and Jurkat cell lysates. This library was generated in Spectronaut using the search algorithm Pulsar. Peptide lengths of 7–52 amino acids and with up to two miscleavages were permitted. The following variable modifications were applied: oxidation (methionine, +15.99 Da), IA-DTB (cysteine, +296.18 Da), acetylation (N-terminus, +42.01 Da), and carbamidomethylation (cysteine, +57.02 Da). All searches were performed against three FASTA files that contained the canonical UniProt human protein sequences, common contaminants[65], and iRT fusion peptides, respectively.

**Data analysis for compound screening.** For analysis of diaPASEF files, standard Biognosys settings were used with minor modifications. In brief, a precursor Q-value cutoff of <0.01 was used with an experiment wide protein Q-value cutoff of <0.01 and a probability cut off for PTM localisation of >0.75. Subsequent analysis was performed using Python. Quantification was performed on the precursor level and intensities associated with equivalent peptides (i.e., only differed in charge state or methionine oxidation state) were summed together to give a single peptide-level intensity. The ability of a compound to complete with the IA-DTB probe was quantified through competition ratios (CR = Intensity$_{DMSO}$/Intensity$_{compound}$) and the significance of the difference between control and compound-treatment samples was calculated using Welch's t-test. The following criteria were used to identify binding events: mean log$_2$(CR) ≥ 1 and -log$_{10}$(p-value) ≥ 1.3. In addition, the peptide was required to be robustly detected without any confounding factors: the peptide must have been detected in at least two of the compound-treated replicates and in ≥90% of all samples in the experiment; the peptide must have a coefficient of variation (CV) of ≤40% in the control samples; the peptide must only have a single DTB modification and, if multiple cleavage forms of the same peptide exist in the dataset, then only the most abundant of these peptides is considered.

In cases where data was acquired at multiple compound concentrations, the mean intensity of each peptide (percent relative to the DMSO control) at varying concentrations (log$_{10}$-transformed) of each compound was fit to a 4-parameter logistic function: $E_{inf} + \frac{E_0 - E_{inf}}{1 + 10^{n(\log_{10}(TE_{50}) - x)}}$, where E$_0$ is the relative peptide intensity when no compound is present (typically the top plateau), E$_{inf}$ is the relative peptide intensity when infinite compound is present (typically the bottom plateau), n is the slope, and TE$_{50}$ is the relative peptide intensity at the midpoint of the curve. Fitting was performed in Python using lmfit[66], with the following bounds: 60 ≤ E$_0$ ≤ 140; E$_{inf}$ ≥ 0; −50 ≤ n ≤ 0; and log$_{10}$(TE$_{50}$) was varied up to 3 log units outside the concentration range tested. The reported errors for best fit values are estimated standard errors calculated by lmfit.

**Global proteomics data analysis.** Raw mass spectrometry data files were analysed using Spectronaut (version 18) with directDIA. The following search parameters were used for directDIA: peptide lengths of 7–52 amino acids with up to two miscleavages were permitted, with one fixed modification (carbamidomethylation; cysteine, +57.02 Da)

and the following variable modifications: oxidation (methionine, +15.99 Da) and acetylation (*N*-terminus, +42.01 Da). All searches were performed against three FASTA files that contained the canonical UniProt human protein sequences, common contaminants[65], and iRT fusion peptides, respectively.

**Protein and residue annotations.** All proteins in the human proteome and their sequences ('one sequence per gene') were obtained from UniProt. The location of cysteine residues within sequences was extracted and annotation of whether these cysteines lie within MS-detectable tryptic sequences was determined through in silico trypsin digestion with the following rules: cleavage after lysine or arginine as long as the next residue is not proline, and permitting peptides with a length of 7–40 amino acids.

Annotations concerning the Illuminating the Druggable Genome (IDG) protein families and target development levels were obtained from the Pharos database. Residue-specific annotations for post-translational modifications and disulfide bonds were obtained from UniProt[67]. Residue-specific pPSE values were obtained from published data[34].

**Chemical and physical properties of the 80-member chloroacetamide library.** Physiochemical properties of the reactive fragments were calculated using LiveDesign (Schrodinger Suite 2023-2). Bemis-Murcko frameworks were assigned manually[68,69].

Principal moment of inertia (PMI) values were calculated using Molecular Operating Environment (MOE: Chemical Computing Group, version: 2019.0101)[70]. A three-dimensional model was first generated for each compound from the SMILES string, by performing a conformational search with the following parameters: force field – MMFF94x; method – stochastic; rejection limit – 200; iteration limit – 10000; amide bond rotation allowed; unconstrained double bond rotation allowed; chair conformations not enforced; not refined with quantum mechanics; root mean squared deviation limit – 0.15; conformation limit – 1. Normalised principal moment of inertia ratios (NPRs) were then calculated from the resulting PMI values.

Molecular similarity was quantified using Morgan fingerprints (radius = 2, bits = 1024) and Tanimoto similarity scores, calculated using RDKit (2022.09.5)[71]. Hierarchical clustering of compounds based on molecular similarity was performed using SciPy with the Ward variance minimisation algorithm[72].

**Fpocket analysis.** To identify which cysteines are located within ligandable pockets, the program Fpocket was applied on monomeric, three-dimensional protein models, as predicted by AlphaFold2[43,44]. Fpocket detects impressions on the protein surface by rolling a series of sphere probes (alpha spheres) with sizes spanning over a specified range of radii. If an alpha sphere touches three atoms of the protein simultaneously, it is placed at that position. By default, a pocket is reported if it contains at least 35 alpha spheres. The range of radii used in our case was 3.0–5.0 Å. We defined a cysteine residue as being located in a pocket if its thiol atom was within 1.5 Å of the closest alpha sphere. The parameters of these calculations were defined empirically, after examining several examples of cysteine liganding events.

### Figure preparation
Figure 1a, b was created using image templates from BioRender.com under the institutional license belonging to the Francis Crick Institute (https://BioRender.com/m32r739).

### Reporting summary
Further information on research design is available in the Nature Portfolio Reporting Summary linked to this article.

## Data availability
The raw mass spectrometry proteomics files and database search results have been deposited at the ProteomeXchange Consortium (http://proteomecentral.proteomexchange.org) via the PRIDE partner repository with data set identifiers PXD054105, PXD054127 and PXD054145[73]. Source Data is provided with this paper as a Source Data file. Source data are provided with this paper.

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

## Acknowledgements

We thank Joanna Kirkpatrick, Toby Baker, and both the Cell Services STP and the Proteomics STP at the Francis Crick Institute for support with this manuscript. We are grateful to Prof. Nick Tomkinson and the University of Strathclyde for their enthusiastic support of a secondment for Harry Wilders into the Crick-GSK LinkLabs during the course of his PhD. This work was supported by the Francis Crick Institute, which receives its core funding from Cancer Research UK (CC2075, CC2000) to K.R. and S.B., the UK Medical Research Council (CC2075, CC2000) to K.R. and S.B., and the Wellcome Trust (CC2075, CC2000) to K.R. and S.B. This project has been funded by a Prosperity Partnership grant from the Engineering and Physical Sciences Research Council (EPSRC), EP/V038028/1 to A.P., D.H., S.B., J.M.S., K.R. and J.B. We also thank GSK for its commitment to support fundamental discovery research through the initial establishment of the Crick-GSK LinkLabs partnership and its contribution to the EPSRC Prosperity Partnership grant.

## Author contributions

G.B. carried out method development, proteomics experiments, and initial data analysis; E.C. carried out all downstream data analysis and visualisation; A.V. assisted in initial method development and sample preparation; W.M.C., H.W., A.v.d.Z. and J.P. assisted in compound selection and annotation; I.R. performed Fpocket analysis; L.N. developed additional scripts for data analysis; P.C. synthesised chemical probes; A.P., D.H., S.B and J.M.S. assisted in project supervision; G.B., E.C., K.R. and J.B. designed the project and wrote the paper with input from all authors. All authors have seen and approved the manuscript.

## Funding

## Competing interests

The authors declare no competing interests.
