## [Transparent Peer Review file · Nature Communications]

Robust proteome profiling of cysteine-reactive fragments using label-free chemoproteomics

Corresponding Author: Dr Jacob Bush

Version 0:

Reviewer comments:

Reviewer #1

(Remarks to the Author)

In the manuscript titled "Robust proteome profiling of cysteine-reactive fragments using label-free chemoproteomic", Biggs et al describe a plate-based label free quantitation (LFQ)/data independent acquisition (DIA) workflow and subsequently deploy this workflow to profile 80 electrophilic fragments in lysates. Using their DIA method, they achieve coverage of ~23K cysteines over 21 min gradients. From their screen of electrophilic fragments, they identify ~400 ligand-protein interactions and perform dose responses and preliminary hit expansion.

Overall, there is much to commend on this study. It is clear that the HT-LFQ workflow presented is highly robust and amendable to industrial-scale to proteome-wide screens. Further, experiments are rigorously conducted and thoughtfully analyzed, this point cannot be emphasized enough. I expect this manuscript and workflow will be of high interest to those pursuing LFQ screens and other chemoproteomic endeavors. However, despite the praise of the technical rigor of this work, it represents a somewhat niche and incremental technical advancement in the field, as DIA-based cysteine profiling methods have been published before and there are also highly robust DDA methods available. Further, this work does not establish any new insights or disclose any novel discoveries enabled by the workflow. Finally, there are instances where the authors contrast their work to other methods, and in doing so, are somewhat misleading about their advances. I expand on these points below.

Considering this, I think this work is not appropriate for publication in Nature Communications but is well-suited for a specialized technical journal, for example, one devoted to proteomics.

- The integration of DIA workflows with cysteine profiling has been reported, for the first time, previously by Chu Wang's lab (J. Am. Chem., 2022, 144, p901). In this work, in addition to developing the DIA method they call DIA-ABPP to profile cysteine reactivity using an iodoacetamide probe, they demonstrate its usage in 2 separate applications, including: 1) profiling covalent ligands proteome-wide, 2) profiling the circadian cysteinome. Interestingly enough, though Biggs et al cite this critical precedence, they do not discuss it nor elaborate how their work is differentiated, which comes off as being intentional. One major differentiation might be that their optimized method compatible includes plate-based preps and has increased coverage, however such technical advancements have also been reported elsewhere (Nat Biotech, 2021, p630, Cell Chem Bio, 2024, p565, and others). The fact that this general method has been reported, benchmarked and used in multiple applications previously, dilutes the novelty.

- Throughout the manuscript, the authors rightly contrast their method with state-of-the-art DDA methods (Nat Biotech, 2021, p630, Cell Chem Bio, 2024, p565, and others). However, it is unclear what the true improvement their method brings to the table. It seems like their main stated improvement is that their HT-LFQ method can screen compounds faster, but in reality, this is actually not true. In fact, they are somewhat misleading in this regard. For example, they consistently contrast run times as being a major advantage, suggesting they can screen more compounds and do things like dose-responses. Yes, their gradient is 21 min long and recent TMT/DIA methods they cite are 3hr long, however, those methods can measure 18 conditions (e.g. ligands) quantitatively at once while the DIA method can do only 1 condition. So DDA method can achieve 60 samples/day, however the TMT cysteine profiling method they cite (Cell Chem Bio, 2024, p565) can actually achieve 144 samples/day in 18-plex experiments. So it would seem their method is actually quite inferior, at least in terms of throughput. It should be noted that the cysteine coverage of their workflow is also slightly less, where the TMT method (Cell Chem Bio, 2024, p565) reports 38,450 quantified cysteines, Biggs et al reports a combined 35,000. It is clear that there are other

advantages, beyond throughput, of using DIA methods, however, as mentioned above, DIA-ABPP has been previously reported (J. Am. Chem., 2022, 144, p901).

- Finally, the authors did not really demonstrate the value of their new workflow. As mentioned in the previous point, it is not clear that their throughput is better than previously reported DDA cysteine profiling methods. Additionally, their screening of fragments did not provide any new insights into cysteine ligandability, it did not yield new chemical probes for proteins, and it did not reveal any new biology, for example. As the authors are aware, there is a high density of literature around cysteine-reactive ligand profiling, and in order to demonstrate the potential impact of their workflow, clear discoveries enabled by it should be presented.

Reviewer #2

(Remarks to the Author)

The authors are presenting a platform for a global chemoproteomics mapping of binding sites to cysteine-reactive fragments using DIA. Such mappings are widely used in drug discovery, and substantial improvements to the technology are very likely to have a large impact on a wide range of these applications. The field has been dominated by using isobaric labeling for quantification. These mappings were originally driven by Cravatt, but Gygi published a seminal paper on pushing the technology in Nat Biotech in 2021 (PMID: 33398154). I think this is the reference paper for the field and, therefore, for the here discussed study by Biggs et al. Biggs et al. are referencing the paper.

Biggs et al. are mapping the binding sites of 80 fragments (less than the 285 Gygi has done in 2021) across two cell lines (3 for Gygi). The new aspect of the paper is the use of DIA (on a Bruker timsTOF Pro 2 (launched in 2021)). The key number is that the authors are mapping 23k cysteine sites in a 21 min run. This is an impressive number (more below). The authors lead us through the method development and some general data analysis of the screening data, they then show some concentration titration experiments for selected ligands/targets. The follow-up experiment results are not overwhelmingly exciting, but that is also not the main message of the paper. All experiments are solidly done, the paper is well-written, and the figures are clear.

DIA is currently the dominating technology in mass spec, and an evaluation against isobaric labeling for fragment ligand mapping is highly important and timely. The concept is not new. Yang et al. have published a study on the concept in 2022 in JACS (PMID: 34986311), but the numbers are not comparable (Yang et al. report 9k sites in a 140 min run on a Q Exactive Plus (launched in 2014), also using outdated technology for cysteine peptide pulldown).

The valid comparison is with Gygi's paper. Biggs et al. state that the 23 k sites they get in 21 minutes match the Gygi's approach. Actually, it's quite better. Gygi reports 16.5 k sites from triplicate 210 min runs (TMT16, 40 min per sample across all duplicates, Orbitrap Lumos (launched 2015)). One should note that the gold posts in the field are moving fast: the new Astral mass spec may substantially improve the depth and throughput of the TMT approach (but also that of DIA) and the new TMT32 reagents will cut the TMT time per sample by 2 bringing both methods into the same throughput ballpark. All in all, the here presented DIA method looks good – if most of the identified peptides are quantifiable. And this is my main suggestion to authors. I think the paper will be great for publication in Nat Comm if they can show that most of the 23 k sites can be quantified with high accuracy and reproducibility. The authors are giving a glimpse at such an evaluation in Supp Fig 1 e. I would like to see more: CV distributions in correlation to precursor intensities or S/N ratios, percentages of peptides within defined CV brackets, and a measurement of how low intensities are affecting the measured ratios. This information is essential for anyone who is planning to use the method and should be presented in a main Fig. Gygi is using an S/N cutoff to ensure quantification quality. For Biggs et al., all aspects of overlap between runs, etc., should be reevaluated using potentially required quantification cut-offs.

Minor comments:

- (1) The difference between all cysteine residues and those in tryptic peptides shown in Fig. 1e should be explained a bit more in the Figure legends (definition of tryptic peptide, length missed cleavages).
- (2) I could not identify any useful information in the heatmaps shown in Supp Fig. 1 d and e. I am not sure if it is necessary to show them.
- (3) Supp Fig. 2. It would help to add the total number of cysteine sites detected.
- (4) In the discussion, the authors mention that DIA can be run on an unlimited number of samples, while isobaric labeling experiments are limited in the number of samples that can be analyzed due to limitations in the number of available barcodes. There are ways to get around the barcode number by adding bridge channels, which has been done by many groups. The authors' statement is a little bold, given that Gygi has run many more samples than they did.

Reviewer #3

(Remarks to the Author)

In "Robust proteome profiling of cysteine-reactive fragments using label-free chemoproteomics", Biggs et al. present a high throughput, label-free quantification method for the identification of reactive cysteine residues throughout the human proteome. The method is exceptionally quick and effective, identifying ~23,000 cysteine residues per 21-minute mass spectrometry run. The newly developed method was used to screen 80 reactive fragments activated with chloroacetamide moieties. The results indicated that the method is capable of discovering cysteine targets from a wide variety of proteins at varying degrees of promiscuity. Hits from this study agree with present literature. The broad utility of the method was demonstrated with concentration-response and structure-activity relationship (SAR) analyses. Compounds screened by SAR were also applied to live cells, demonstrating the applicability of this method in vivo systems, complementing the work done in vitro. We believe this work is well suited for Nature Communications and recommend acceptance with minor

revisions. The following points should be considered for revisions:

1. In the introduction, the “ambition of the research community to discover chemical probes for every expressed protein” should be referenced (line 14, pg. 3).
2. Due to the emphasis placed on the high throughput nature of this method, the authors should include an estimated time taken for the sample preparation in the results section.
3. It is unclear why GTSO1-PP1 is included as a top five interaction in the text and in Fig. 2e, but not included in Fig. 3 (pg.8, line 35). We suggest that an explanation is provided, or the volcano plot is added to Fig. 3, or that it is removed from the text and not highlighted in Fig. 2e.
4. The live cell data for the compounds used in SAR (presented in Fig. 5c) largely agrees with the in vitro data, however there are apparent discrepancies in the specificity, as shown by the volcano plots. These discrepancies should be addressed in the discussion or figure caption, and interested readers should be referred to the appropriate supplementary data.

Version 1:

Reviewer comments:

Reviewer #1

(Remarks to the Author)

I appreciate the response from the authors, and their additional clarification regarding comparisons to other works. In addition, as mentioned in my original review, it is clear that the workflow presented is highly robust and amendable to industrial-scale to proteome-wide screens. Further, experiments are rigorous and I expect workflow to be of high interest to those pursuing chemoproteomic LFQ screens.

However, my main concerns are still around novelty and impact. This work is mostly a technical advancement, as DIA-based cysteine profiling methods have been published before (albeit, not comparable to this work) and there are also highly robust DDA methods available. Though there are some advancements, there aren't any new insights or novel discoveries enabled by the workflow. I think it is more appropriate for publication in a specialized technical journal devoted to proteomics

Reviewer #2

(Remarks to the Author)

Looks good overall. There are a few things I would like to be included in the manuscript for publication.

- (i) Supp Fig 1 (e): I would like to see the correlations of CR ratios in addition to the absolute intensity correlation plots. All plot description should contain an n.
- (ii) For all CV plots Supp Fig 1 f/g it is not clear if these are peptides from all datasets or only from the replicate pairs described in Supp Fig 1 (d), again n is missing.
- (iii) As the authors state in the rebuttal letter, and as is described in the method, the authors are using some filter criteria for the ligand screening (peptide must have been detected in at ≥ 2 compound-treated samples and in ≥ 90 % of all samples; peptide CV ≤ 40 % in control samples, only a single DTB). The total number of quantified sites, average per run, overlap between runs, should be given for the dataset upon this filtering. This is important information for the reader and should be mentioned in the main text.

Once this additional information is added to the manuscript, it is suitable for publication in Nat Comm

Reviewer #3

(Remarks to the Author)

The authors' revisions are satisfactory. The manuscript is suitable for publication.

Version 2:

Reviewer comments:

Reviewer #2

(Remarks to the Author)

This looks very good. Thanks to the authors for going through this extra effort. I believe the manuscript is suitable for publication in Nature Communications.

Response to Reviewers

Reviewer #1 (Remarks to the Author):

In the manuscript titled “Robust proteome profiling of cysteine-reactive fragments using label-free chemoproteomic”, Biggs et al describe a plate-based label free quantitation (LFQ)/data independent acquisition (DIA) workflow and subsequently deploy this workflow to profile 80 electrophilic fragments in lysates. Using their DIA method, they achieve coverage of ~23K cysteines over 21min gradients. From their screen of electrophilic fragments, they identify ~400 ligand-protein interactions and perform dose responses and preliminary hit expansion.

Overall, there is much to commend on this study. It is clear that the HT-LFQ workflow presented is highly robust and amendable to industrial-scale to proteome-wide screens. Further, experiments are rigorously conducted and thoughtfully analyzed, this point cannot be emphasized enough. I expect this manuscript and workflow will be of high interest to those pursuing LFQ screens and other chemoproteomic endeavors.

However, despite the praise of the technical rigor of this work, it represents a somewhat niche and incremental technical advancement in the field, as DIA-based cysteine profiling methods have been published before and there are also highly robust DDA methods available.

Further, this work does not establish any new insights or disclose any novel discoveries enabled by the workflow. Finally, there are instances where the authors contrast their work to other methods, and in doing so, are somewhat misleading about their advances. I expand on these points below.

Considering this, I think this work is not appropriate for publication in Nature Communications but is well-suited for a specialized technical journal, for example, one devoted to proteomics.

We thank the Reviewer for these positive comments about the technical rigor of our manuscript, and address the Reviewer's concerns to the significance of the advances with the following points below.

- The integration of DIA workflows with cysteine profiling has been reported, for the first time, previously by Chu Wang's lab (J. Am. Chem, 2022, 144, p901). In this work, in addition to developing the DIA method they call DIA-ABPP to profile cysteine reactivity using an iodoacetamide probe, they demonstrate its usage in 2 separate applications, including: 1) profiling covalent ligands proteome-wide, 2) profiling the circadian cysteinome. Interestingly enough, though Biggs et al cite this critical precedence, they do not discuss it nor elaborate how their work is differentiated, which comes off as being intentional.

We are grateful to the Reviewer for these comments. Throughout our manuscript, we have attempted to avoid direct comparisons with previous literature reports, since the approaches are often not directly comparable. Instead, we have been very intentional about highlighting the key strengths of our platform and actively cited other approaches in order to allow readers to make their own informed assessment. Each previously reported competitive cysteine-based platform (dia-SLC-ABPP, isoTOPP-ABPP, DIA-ABPP and now our own HT-LFQ cysteine profiling) is unique and have their own strengths.

For the DIA-ABPP platform referred to by Reviewer 1 (PMID: 34986311), DIA is combined with an iso-TOPP labelling strategy, whereas our platform combines DIA-PASEF with label free quantification (LFQ).

As highlighted by Reviewer 2, this enabled us to achieve much improved cysteine identifications in comparison to DIA-ABPP: ~23,000 cysteines identified per 21-minute run compared to ~9,000 cysteine sites in a 140-minute run on a Q-Exactive plus. We are confident this is a significant advance in cysteinome depth and highlights the usefulness of our platform.

We draw the Reviewer's attention to Page 12, Line 19-22 in the manuscript where we highlight the advances in cysteinome depth offered by our LFQ-DIA based approach:

"Furthermore, an extra level of peptide separation was performed via trapped ion-mobility using PASEF (on a Bruker timsTOF Pro 2), which provides unrivalled cysteinome depth for DIA-based cysteine profiling (~23,000 cysteines identified per run; ~30,000 per experiment)."

One major differentiation might be that their optimized method compatible includes plate-based preps and has increased coverage, however such technical advancements have also been reported elsewhere (Nat Biotech, 2021, p630, Cell Chem Bio, 2024, p565, and others). The fact that this general method has been reported, benchmarked and used in multiple applications previously, dilutes the novelty.

The Reviewer is correct that the publication in Cell Chemical Biology (PMID: 38118439), reported a related plate-based workflow, which we have referenced accordingly. We believe the plate-based platform reported in our manuscript provides a useful alternative methodology for the community and has the following benefits:

- Our workflow is fully plate-based, from lysate treatment all the way to loading on the mass spectrometer, whereas in the reported Cell Chem Bio paper, once the TMT channels are combined, the enrichment and desalting is performed in Eppendorf tubes.
- Our workflow for protein clean-up consists of a low-cost SP4 glass bead workflow, which allows for consistent protein recovery and efficient removal of excess small molecules and detergents.
- The SP4 workflow is a filter-plate based workflow ensuring that no beads end up in mass spectrometry ready samples.

To better highlight these differences, we highlight the additional text added to the manuscript: (Page 5, Line 16-18):

"In this workflow, samples remain in 96-well plate format from compound treatment through to mass spectrometer injection, and a single experimentalist can readily prepare 384 samples (4 plates) in 2-3 days.

- Throughout the manuscript, the authors rightly contrast their method with state-of-the-art DDA methods (Nat Biotech, 2021, p630, Cell Chem Bio, 2024, p565, and others). However, it is unclear what the true improvement their method brings to the table. It seems like their main stated improvement is that their HT-LFQ method can screen compounds faster, but in reality, this is actually not true. In fact, they are somewhat misleading in this regard. For example, they consistently contrast run times as being a major advantage, suggesting they can screen more compounds and do things like dose-responses. Yes, their gradient is 21 min long and recent TMT/DIA methods they cite are 3hr long, however, those methods can measure 18 conditions (e.g. ligands) quantitatively at once while the DIA method can do only 1 condition. So DDA method can achieve 60 samples/day, however the TMT cysteine profiling method they cite (Cell Chem Bio, 2024, p565) can actually achieve 144 samples/day in 18-plex experiments. So it would seem

their method is actually quite inferior, at least in terms of throughput. It should be noted that the cysteine coverage of their workflow is also slightly less, where the TMT method (Cell Chem Bio, 2024, p565) reports 38,450 quantified cysteines, Bigg's et al reports a combined 35,000. It is clear that there are other advantages, beyond throughput, of using DIA methods, however, as mentioned above, DIA-ABPP has been previously reported (J. Am. Chem, 2022, 144, p901).

We thank the Reviewer for highlighting that we were not sufficiently clear in pointing out the advantages of our new method. To express these advantages more explicitly we have edited the manuscript to clarify any statements we have made where we compare our HT-LFQ platform with previously reported DDA-TMT platforms from the Gygi Lab (2021) (PMID: 33398154) and the Yang Lab (2024) (PMID: 38118439).

First, we feel the true improvements this method brings in comparison to these DDA-TMT platforms is the combination of high cysteinome depth per-run, coupled with high data completeness between different runs. Using our label-free DIA platform we quantify ~23,000 cysteine containing peptides using 21-minute chromatographic gradients per compound. This compares favourably (with respect to cysteinome depth per run) with the Gygi (2021) publication (~16,500 cysteine sites, with 18-minute per compound) and the Yang (2024) publication (~19,000 cysteine sites, with 11.5-minutes per compound). A key advantage afforded by our label free DIA platform is the high overlap of cysteine sites between different control samples (i.e. data completeness), as demonstrated by Figure 1c – 1d and reported in Page 6, Line 2-4.

To further compare, we report that ~54% of cysteines are present in 16/16 separate mass spectrometry runs, whereas the Yang (2024) publication reports ~35% of cysteines are present in 12/12 separate mass spectrometry runs. This improved data completeness increases confidence in the liganding interactions that are reported.

Second, we apologise for any misunderstanding we have caused in comparing our throughput with DDA-TMT based methods. As pointed out by the Reviewer, and discussed in the previous paragraph, our time/sample is comparable to the original Gygi (2021) publication and slightly slower than the recent Yang (2024) publication. In order to address this and clarify our statement, we have rephrased the sentence in which we compare throughput of our method with other reported approaches (Page 4, Line 7-9):

“The method offers sensitivity and throughput that compares favourably with reported methods to date, and offers much improved reproducibility and data completeness.”

Which now reads,

“The method offers sensitivity and cysteine coverage per run that compares favourably with reported methods to date, and offers much improved reproducibility and data completeness.”

- Finally, the authors did not really demonstrate the value of their new workflow. As mentioned in the previous point, it is not clear that their throughput is better than previously reported DDA cysteine profiling methods. Additionally, their screening of fragments did not provide any new insights into cysteine ligandability, it did not yield new chemical probes for proteins, and it did not reveal any new biology, for example. As the authors are aware, there is a high density of literature around cysteine-reactive ligand

profiling, and in order to demonstrate the potential impact of their workflow, clear discoveries enabled by it should be presented.

We hope that our replies to the previous points raised by this Reviewer have helped to more clearly demonstrate the value of our new label free DIA workflow, specifically in terms of cysteine depth per run and data completeness. Furthermore, we believe that our study convincingly shows that this platform is capable of screening reactive fragment libraries by chemoproteomics, of which there have been few examples to date. We have found ligands for >400 cysteine sites and we believe that both these robust liganding interactions will be a useful resource for the chemical biology and drug discovery community.

Reviewer #2 (Remarks to the Author):

The authors are presenting a platform for a global chemoproteomics mapping of binding sites to cysteine-reactive fragments using DIA. Such mappings are widely used in drug discovery, and substantial improvements to the technology are very likely to have a large impact on a wide range of these applications. The field has been dominated by using isobaric labeling for quantification. These mappings were originally driven by Cravatt, but Gygi published a seminal paper on pushing the technology in Nat Biotech in 2021 (PMID: 33398154). I think this is the reference paper for the field and, therefore, for the here discussed study by Biggs et al. Biggs et al. are referencing the paper.

Biggs et al. are mapping the binding sites of 80 fragments (less than the 285 Gygi has done in 2021) across two cell lines (3 for Gygi). The new aspect of the paper is the use of DIA (on a Bruker timsTOF Pro 2 (launched in 2021)). The key number is that the authors are mapping 23k cysteine sites in a 21 min run. This is an impressive number (more below). The authors lead us through the method development and some general data analysis of the screening data, they then show some concentration titration experiments for selected ligands/targets. The follow-up experiment results are not overwhelmingly exciting, but that is also not the main message of the paper. All experiments are solidly done, the paper is well-written, and the figures are clear. DIA is currently the dominating technology in mass spec, and an evaluation against isobaric labeling for fragment ligand mapping is highly important and timely. The concept is not new. Yang et al. have published a study on the concept in 2022 in JACS (PMID: 34986311), but the numbers are not comparable (Yang et al. report 9k sites in a 140 min run on a Q Exactive Plus (launched in 2014), also using outdated technology for cysteine peptide pulldown).

We thank the Reviewer for their supportive and enthusiastic assessment of our manuscript and the advances in the chemoproteomics method described therein.

The valid comparison is with Gygi's paper. Biggs et al. state that the 23 k sites they get in 21 minutes match the Gygi's approach. Actually, it's quite better. Gygi reports 16.5 k sites from triplicate 210 min runs (TMT16, 40 min per sample across all duplicates, Orbitrap Lumos (launched 2015)). One should note that the gold posts in the field are moving fast: the new Astral mass spec may substantially improve the depth and throughput of the TMT approach (but also that of DIA) and the new TMT32 reagents will cut the TMT time per sample by 2 bringing both methods into the same throughput ballpark.

We thank this Reviewer for the favourable assessment of our new workflow compared to the DDA-TMT methods reported by the Gygi Lab. We also agree, and are very excited by how new developments in mass spectrometry and data analysis platforms will expand cysteinome coverage. In addition to the Thermo Astral mass spectrometer, there have also been key developments in the timsTOF series of mass spectrometers, including the timsTOF HT (released 2022) and the timsTOF Ultra 2 (released 2024) which will likely lead to increased coverage and reduce the input requirement.

All in all, the here presented DIA method looks good – if most of the identified peptides are quantifiable. And this is my main suggestion to authors. I think the paper will be great for publication in Nat Comm if they can show that most of the 23 k sites can be quantified with high accuracy and reproducibility. The authors are giving a glimpse at such an evaluation in Supp Fig 1 e. I would like to see more: CV distributions in correlation to precursor intensities or S/N ratios, percentages of peptides within defined CV brackets, and a measurement of how low intensities are affecting the measured ratios. This information is essential for anyone who is planning to use the method and should be presented in a main Fig. Gygi is using an S/N cutoff to ensure quantification quality. For Biggs et al., all aspects of overlap between runs, etc., should be reevaluated using potentially required quantification cut-offs.

This is an excellent suggestion, which we have addressed by adding further figures, parts (f) and (g) to Supplementary Fig 1 (shown below). The panel titled “Reproducibility of peptide intensities measured” demonstrates the consistency in quantification for the peptides we measure. In part (e) we show the high Pearson correlation between any two given replicates (median = 0.96) from 16 separate mass spectrometry samples. In part (f) we further address the reviewer’s comments by showing the CV distribution of every peptide that has been detected in at least 2 replicates. The median CV for peptide level quantification is 24.8. In part (g) we demonstrate how the CV distributions for these peptides correlate with measured intensity. As predicted, lower intensity peptides have a higher CV distribution, and higher peptide intensity have a lower CV distribution. We have edited the following text to the manuscript to address these comments (Page 6, Line 7-9):

“Furthermore, excellent reproducibility of the peptide intensities was observed, with a median Pearson correlation between replicates of 0.96 (Supplementary Fig. 1e), and a median coefficient of variation of 24.8% (Supplementary Fig. 1f-g).”

In fragment screens where we report liganding interactions, we use more stringent QC thresholds than are used typically in the field. As reported in the Methods, the peptide has to be detected in at least two of the compound-treated replicates; in $\geq 90\%$ of all samples in the experiment and the peptide must have a coefficient of variation (CV) of $\leq 40\%$ in the control samples. This removes peptides for which there is a poor quantification and increases confidence in the liganding interactions reported.

Reproducibility of peptide intensities measured

Minor comments:

(1) The difference between all cysteine residues and those in tryptic peptides shown in Fig. 1e should be explained a bit more in the Figure legends (definition of tryptic peptide, length missed cleavages).

We thank the reviewer for spotting this and have added the following text to the figure legend for Fig 1:

“Tryptic peptides were classified as being detectable if they were 7-40 residues (not considering missed cleavages).”

Additionally, a more detailed description of the *in silico* digestion that we performed is described in Methods (Page 18, Line 26-30).

(2) I could not identify any useful information in the heatmaps shown in Supp Fig. 1 d and e. I am not sure if it is necessary to show them.

We would prefer to keep the heatmaps shown in Supplementary Fig. 1 because, we believe that the heatmaps aid in demonstrating the high data completeness achieved across 16 replicates and the reproducibility in peptide quantification.

(3) Supp Fig. 2. It would help to add the total number of cysteine sites detected.

We thank the reviewer for spotting this and have added the information in Supplementary Figure 2.

(4) In the discussion, the authors mention that DIA can be run on an unlimited number of samples, while isobaric labeling experiments are limited in the number of samples that can be analyzed due to limitations

in the number of available barcodes. There are ways to get around the barcode number by adding bridge channels, which has been done by many groups. The authors' statement is a little bold, given that Gygi has run many more samples than they did.

We agree with this reviewer that our statement was too bold and have removed the following sentence (Page 12, Line 16-17):

“In contrast with TMT-based experiments, where quantitative comparison is performed in a single mass spectrometry run and is limited by the number of multiplexed samples, label-free quantification allows for comparison of peptide quantities between an unlimited number of samples.”

This has been replaced with the following sentence, to remove the comparison to TMT-DDA:

“Our LFQ based chemoproteomics platform allows for comparison of peptide quantities between an unlimited number of samples.”

Reviewer #3 (Remarks to the Author):

In “Robust proteome profiling of cysteine-reactive fragments using label-free chemoproteomics”, Biggs et al. present a high throughput, label-free quantification method for the identification of reactive cysteine residues throughout the human proteome. The method is exceptionally quick and effective, identifying ~23,000 cysteine residues per 21-minute mass spectrometry run. The newly developed method was used to screen 80 reactive fragments activated with chloroacetamide moieties. The results indicated that the method is capable of discovering cysteine targets from a wide variety of proteins at varying degrees of promiscuity. Hits from this study agree with present literature. The broad utility of the method was demonstrated with concentration-response and structure-activity relationship (SAR) analyses. Compounds screened by SAR were also applied to live cells, demonstrating the applicability of this method in vivo systems, complementing the work done in vitro. We believe this work is well suited for Nature Communications and recommend acceptance with minor revisions.

We are grateful to this Reviewer for their positive assessment and recognition of the broad utility of this chemoproteomics platform.

The following points should be considered for revisions:

1. In the introduction, the “ambition of the research community to discover chemical probes for every expressed protein” should be referenced (line 14, pg. 3).

We agree with this point, and have added a reference to the Target 2035 initiative. These are covered by references 3, 7 and 8.

2. Due to the emphasis placed on the high throughput nature of this method, the authors should include an estimated time taken for the sample preparation in the results section.

To emphasise the high throughput nature of our sample preparation platform we have added the following sentence to the Results section of the manuscript:

“In this workflow, samples remain in 96-well plate format from compound treatment through to mass spectrometer injection, and a single experimentalist can readily prepare 384 samples (4 plates) in 2-3 days.”

3. It is unclear why GTSO1-PP1 is included as a top five interaction in the text and in Fig. 2e, but not included in Fig. 3 (pg.8, line 35). We suggest that an explanation is provided, or the volcano plot is added to Fig. 3, or that it is removed from the text and not highlighted in Fig. 2e.

We thank the reviewer for this point and opportunity to clarify this. We have added the following sentence to the manuscript (Page 9, Line 1-2):

“The interaction between GTSO1 Cys32 and PP1 was deprioritised for follow up, as it has been liganded by covalent fragments in multiple other datasets.”

4. The live cell data for the compounds used in SAR (presented in Fig. 5c) largely agrees with the in vitro data, however there are apparent discrepancies in the specificity, as shown by the volcano plots. These discrepancies should be addressed in the discussion or figure caption, and interested readers should be referred to the appropriate supplementary data.

We thank the reviewer for this suggestion and we have modified the text and added Supplementary Fig 9c to address the small discrepancies that appear between live-cell and lysate-based experiments (Page 11, Line 2-5).

“Importantly, we confirmed the binding of PP48 to MOB4 Cys134, NIT1 Cys203, and NIT2 Cys153, with additional off-targets also identified (e.g., SORD Cys45), potentially due to the increased incubation time employed for the live cell treatment (1 h vs 2 h) or increased temperature (room temperature vs 37°C) (Supplementary Fig. 9c).”

REVIEWER COMMENTS

Reviewer #2 (Remarks to the Author):

Looks good overall. There are a few things I would like to be included in the manuscript for publication.

We thank the reviewer for these comments and the detailed review of our manuscript.

(i) Supp Fig 1 (e): I would like to see the correlations of CR ratios in addition to the absolute intensity correlation plots. All plot description should contain an n.

In this supplementary figure we report the correlation of intensity values from 16 different control samples. This doesn't include any compound treated samples where competition ratios (CRs) are calculated. We appreciate the reviewer's suggestion to include some plots to summarise the consistency of CR values.

We do not calculate CR on a per replicate basis – but rather using a mean of peptide intensity in control samples ($n \geq 12$) and a mean of peptide intensity in treated samples ($n \geq 4$). We then calculate p-values using a Welch's T-test and perform careful filtering to ensure robust hit calling, which we have detailed in Supplementary Table 2 (added and explained below). Thus, we are unable to plot the correlation of CR between replicates. Instead, in order to address the reviews suggestion, we have calculated the CV of our CRs using propagation of standard deviation error of the two means (treatment and controls). We have done this for both liganded data (meanCR > 2, $-\log(p\text{-value}) > 1.3$) and all data. See the two histogram plots below.

We do not feel these two plots add value to the manuscript, since the peptide intensity is reduced for liganded sites, leading to an increase in the CV, which is reflective of an increased contribution of noise and not a reduction in confidence of liganding. In order to robustly determine liganding events we have used other (in our opinion more suitable) statistical metrics as outlined in the manuscript, for example peptide presence in $\geq 90\%$ of all samples, CV filter on the control samples, and a p-value calculation.

We would prefer not to add these plots to the manuscript for reasons described above but will do so if it is felt necessary for publication in Supplementary Figure 6, which summarises both 80 compound screens in HEK and Jurkat lysate.

Figure Legend: The coefficient of variation (CV) distribution of all competition ratios that pass through all filters (left) and those we define as liganded (right) ($\log_2\text{meanCR} > 1$, $-\log(p\text{-value}) > 1.3$). These CV values are propagated from the standard deviations of treatment and control peptides.

We thank the reviewer for highlighting that it would be useful to add a value for n (i.e. the number of peptides/points on a given plot) to Figure 1(e). We have added n numbers to the Supplementary figure legends for Figure 1(e), and also (f) and (g) as reported below.

(ii) For all CV plots Supp Fig 1 f/g it is not clear if these are peptides from all datasets or only from the replicate pairs described in Supp Fig 1 (d), again n is missing.

These CV plots are intensity correlations between peptides that have been identified in at least two replicates, n=28954. We have edited the figure legend to clarify this and added the n number accordingly.

(iii) As the authors state in the rebuttal letter, and as is described in the method, the authors are using some filter criteria for the ligand screening (peptide must have been detected in at ≥ 2 compound-treated

samples and in $\geq 90\%$ of all samples; peptide CV $\leq 40\%$ in control samples, only a single DTB. The total number of quantified sites, average per run, overlap between runs, should be given for the dataset upon this filtering. This is important information for the reader and should be mentioned in the main text.

We thank the reviewer for suggesting we add this additional information. As previously discussed, when highlighting the most robust liganding events we applied strict filters. To capture the effect of these filters on the number of quantified sites we have added a Supplementary Table 2. The average number and overlap of peptides between runs are both very high since we have filtered for peptides in $\geq 90\%$ of all samples. We have edited the main text according, pointing directly to Supplementary Table 2 (shown below).

“We defined liganded peptides as those with statistically significant competition of at least 50% ($\log_2(\text{CR}) \geq 1.0$, $-\log_{10}(\text{p-value}) \geq 1.3$).”

To this on Page 7 Ln 24-26:

“To focus on the most robust liganding events in each screen, we performed strict peptide filtering and have defined liganded peptides as those with statistically significant competition of at least 50% ($\log_2(\text{CR}) \geq 1.0$, $-\log_{10}(\text{p-value}) \geq 1.3$). The filters we have applied are described in detail in the experimental methods and the results on peptide numbers in **Supplementary Table 2.**”

Supplementary Table 2: Unique peptide counts after filtering steps for each screen that was performed. The numbers quoted represent the number of peptides that pass each filtering step (1-5) when these are applied sequentially.

	Initial screen: HEK293T lysate, 80 compounds	Initial screen: Jurkat lysate, 80 compounds	Validation: HEK293T lysate, 8 compounds	Lysate SAR: HEK293T lysate, 8 compounds	Live cell SAR: HEK293T live cells (2 h), 4 compounds
Total samples	335	328	334	191	48
Total peptides	31107	29818	29096	27954	28664
1. Single DTB modification	30156	29034	28190	27198	28349
2. In $\geq 75\%$ control samples	19037	18895	16123	15325	18175
3. Control sample CV $< 40\%$	15818	12200	11783	13215	16205
4. Most abundant/only cleavage product	14658	11231	10892	12174	15145
5. Present in $\geq 90\%$ of all samples	13631	10321	9552	11174	14094

Once this additional information is added to the manuscript, it is suitable for publication in Nat Comm